**Aerosol Vertical Distribution and Interactions with Land/Sea Breezes over the Eastern**
**Coast of the Red Sea from LIDAR Data and High-resolution WRF-Chem Simulations**
Sagar P. Parajuli[1*], Georgiy L. Stenchikov[1], Alexander Ukhov[1], Illia Shevchenko[1], Oleg
Dubovik[2], and Anton Lopatin[3]
[1]King Abdullah University of Science and Technology, Thuwal, Saudi Arabia
[2]Univ. Lille, CNRS, UMR 8518 - LOA - Laboratoire d'Optique Atmosphérique, F-59000 Lille,
France
[3]GRASP-SAS, Remote Sensing Developments, Université de Lille, Villeneuve D' ASCQ,
59655, France
[*]Corresponding Author, E-mail: psagar@utexas.edu

## Abstract

With advances in modeling approaches and the application of satellite and ground-based data in dust-related research, our understanding of the dust cycle has significantly improved in recent decades. However, two aspects of the dust cycle, namely the vertical profiles and diurnal cycles, are not yet adequately understood, mainly due to the sparsity of direct observations. Measurements of backscattering caused by atmospheric aerosols have been ongoing since 2014 at the King Abdullah University of Science and Technology (KAUST) campus using a micro-pulse Lidar (MPL) with a high temporal resolution. KAUST is located on the east coast of the Red Sea, and currently hosts the only operating LIDAR system in the Arabian Peninsula. We use the data from the MPL together with other collocated observations and high-resolution simulations (with 1.33 km grid spacing) from Weather Research and Forecasting model coupled with Chemistry (WRF-Chem) to study the following three aspects of dust over the Red Sea coastal plains. Firstly, we compare the model simulated surface winds, aerosol optical depth (AOD), and aerosol size distributions with observations, and evaluate the model performance in representing a typical large-scale dust event over the study site. Secondly, we investigate the vertical profiles of aerosol extinction and concentration in terms of their seasonal and diurnal variability. Thirdly, we explore the interactions between dust aerosols and land/sea breezes, which are the most influential components of the local diurnal circulation in the region.

The WRF-Chem model successfully reproduced the diurnal profile of surface wind speed, AOD, and dust size distributions over the study area compared to observations. The model also captured the onset, demise, and height of a large-scale dust event that occurred in 2015, as compared to the LIDAR data. The vertical profiles of aerosol extinction in different seasons were largely consistent between the MPL data and WRF-Chem simulations along with key observations and reanalyses used in this study. We found a substantial variation in the vertical profile of aerosols in different seasons, and between daytime and nighttime, as revealed by the MPL data. The MPL data also identified a prominent dust layer at ~5–7 km during the nighttime, which likely represents the long-range transported dust brought to the site by the easterly flow from remote inland deserts.

The sea breeze circulation was much deeper (~2 km) than the land breeze circulation (~1 km), but both breeze systems prominently affected the distribution of dust aerosols over the study site. We observed that sea breezes push the dust aerosols upwards along the western slope of the Sarawat Mountains. These sea breezes eventually collide with the dust-laden northeasterly trade winds coming from nearby inland deserts, thus causing elevated dust maxima at a height of ~1.5 km above sea level over the mountains. Moreover, the sea and land breezes intensify dust emissions from the coastal region during the daytime and nighttime, respectively. Our study, although focused on a particular region, has broader environmental implications as it highlights how aerosols and dust emissions from the coastal plains can affect the Red Sea climate and marine habitats.

## 1. Introduction

Dust aerosols, which mainly originate from natural deserts and disturbed soils such as agricultural areas, have implications for air quality (Prospero, 1999; Parajuli et al., 2019) and the Earth's climate (Sokolik and Toon, 1996; Mahowald et al., 2006; Prakash et al., 2014; Bangalath and Stenchikov, 2015; Kalenderski and Stenchikov, 2016; Di Biagio et al., 2017). The Arabian Peninsula represents a key area within the global dust belt where significant dust emissions occur in all seasons. However, the spatio-temporal characteristics of dust emissions in the region have not yet been fully described, partly because of the sparsity of observations. Although our understanding of the dust cycle and the related physical processes has substantially improved in recent decades (Shao et al., 2011), in the present context, two aspects of dust aerosol dynamics remain the least explored: the vertical structure and the diurnal cycle. Understanding the vertical structure is important because the vertical distribution of aerosols affects the radiative budget (Johnson et al., 2008; Osipov et al., 2015) and surface air quality (Chin et al., 2007; Wang et al., 2010; Ukhov et al., 2020a). Moreover, understanding the diurnal cycle of aerosols is important because aerosols scatter and absorb radiation (Sokolik and Toon, 1998; Di Biagio et al., 2017), which ultimately affects the land and sea breezes in coastal areas. Land and sea breezes, which are the key diurnal-scale atmospheric processes over the Red Sea coastal plain, can also affect the distribution and transport of aerosols (Khan et al., 2015) and their composition (Fernández-Camacho et al., 2010; Derimian et al., 2017).

The vertical distribution of aerosols in the atmosphere has been studied for decades using LIDAR measurements from several ground-based sites, aircraft, and satellite platforms, covering different regions across the globe. Several satellites are equipped with LIDAR to measure the vertical distribution of aerosols. Lidar In-space Technology Experiment (LITE) was the first space lidar launched by NASA in 1994 onboard the Space Shuttle, providing a quick snapshot of aerosols and clouds in the atmosphere on a global scale (Winker et al., 1996). LITE was followed by the Geoscience Laser Altimeter System (GLAS) containing a 532-nm LIDAR, as part of the Ice, Cloud and Land Elevation Satellite (ICESat) mission, which covered the polar regions (Abshire et al., 2005). Cloud-Aerosol Lidar with Orthogonal Polarization (CALIOP) onboard CALIPSO (Cloud-Aerosol Lidar and Infrared Pathfinder Satellite Observations) currently observes aerosol and clouds globally during both the day and night portion of the orbit with a 16-day repeat cycle since 2006 (Winker et al., 2013). Apart from satellites, several field experiments have also been conducted using LIDAR to measure the vertical distribution of aerosols. The Indian Ocean Experiment (INDOEX) field campaign (Collins et al., 2001; Rasch et al., 2001; Welton et al., 2002b) took place in 1999 over the Indian Ocean, Arabian Sea, and the Bay of Bengal, in which an MPL system together with other instruments measured aerosol distribution in the troposphere. Similarly, an MPL system was employed in the Second Aerosol Characterization Experiment (ACE-2) in 1997 over Tenerife, Canary Islands, to understand the vertical distribution of dust/aerosols transported from North Africa and Europe to the Atlantic Ocean (Welton et al., 2000; Ansmann et al., 2002). African Monsoon Multidisciplinary Analysis (AMMA), one of the largest international projects ever carried out in Africa, also measured aerosol vertical distribution using multiple LIDAR systems for a short period in 2006 (Heese and Wiegner, 2008; Lebel et al., 2010). Currently, several other coordinated LIDAR networks are

operating in different regions. They include the European Aerosol Research Lidar Network
EARLINET (Pappalardo et al., 2014), German Aerosol Lidar Network (Boesenberg et al., 2001),
the Latin American Lidar Network LALINET (Guerrero-Rascado et al., 2016), the Asian dust
and aerosol lidar observation network AD-Net (Shimizu et al., 2016), and the Commonwealth of
Independent States Lidar Network CIS-LiNet (Chaikovsky et al., 2006).
A micro-pulse LIDAR (MPL) has been operating at King Abdullah University of Science and
Technology (KAUST), Thuwal, Saudi Arabia (22.3° N, 39.1° E), since 2014. This LIDAR is
collocated with the KAUST AERONET (Aerosol Robotic Network) station. The KAUST MPL
site is a part of the Micro-Pulse Lidar Network (MPLNET), maintained by the NASA Goddard
Space Flight Center (GSFC) (Welton et al., 2001; Welton et al., 2002a). KAUST hosts the only
LIDAR site on the Red Sea coast, and its colocation with the AERONET station facilitates the
retrieval of the vertical profile of aerosols more accurately. Stations that measure a range of
parameters of interest for dust-related research (including dust deposition rate, vertical profile,
near-surface concentration, and spectral optical depth) are rare across the global dust belt. In
addition to the LIDAR and AERONET station, KAUST also has a meteorological station that
measures wind speed, air temperature, and incoming short-wave and long-wave radiative fluxes.
These collocated data provide an opportunity to get a more complete picture of dust emissions
and transport in the region.
The study site frequently experiences large-scale dust events. The satellite and ground-based
observations such as AERONET have some limitations, because of which they are likely to miss
some important details of these dust events. For example, many large-scale dust events are
accompanied by cloud cover, which restricts the retrieval of aerosol optical properties in the
visible bands (Fernández et al., 2019). Extreme dust events are nonetheless important from a
research perspective because they provide an opportunity to understand the associated physical
processes. AERONET stations and passive satellite sensors are further limited because they
cannot retrieve aerosol properties during the night. LIDARs help to overcome these limitations
because they provide high-frequency measurements, even at night. Furthermore, LIDAR signals
can penetrate thin and multilayer clouds, which are usually overlooked by passive satellite
sensors (Winker et al., 1996; Winker et al., 2009), thus improving the detection of aerosol layers
at different altitudes. Therefore, LIDAR data are essential for understanding the diurnal
variability of aerosols and their climatic effect.
The location of the Red Sea between the two key dust source regions of North Africa and the
Arabian Peninsula provides a unique opportunity to understand the multi-faceted aspects of
aerosol-climate interactions that occur in the region. KAUST is located on the eastern coast of
the Red Sea, and dust is indeed the dominant aerosol type in this region (Prakash et al., 2014;
Kalenderski and Stenchikov, 2016).  The sea and land breezes that occur during the day and
night, respectively, are the dominant drivers of local air mass circulations (Jiang et al., 2009).
Sea breezes facilitate the transport of moisture inland and contribute to the formation of cumulus
clouds and mesoscale convection (Davis et al., 2019). The land and sea breezes can themselves
also generate dust emissions from the coastal regions (e.g., Crouvi et al., 2017), and also interact
with atmospheric dust aerosols in multiple ways.

In this study, we attempt to understand the vertical and diurnal profiles of aerosols over the
eastern coast of the Red Sea. We use our multiple collocated datasets collected at KAUST to
shed light on the various facets of local-scale dust-climate interactions in the region. Since land
and sea breezes are fine-scale features modulated by local topography, high-resolution
simulations are essential to resolve these circulations. Therefore, we conduct high-resolution
simulations (with 1.33 km grid spacing) using WRF-Chem that interactively accounts for aerosol
generation, transport, and deposition to understand the nature of these circulations and their
interaction with aerosols. In summary, we aim to answer the following specific research
questions:

1. How do the model simulations represent the vertical distribution of aerosols over the
   study site?
2. How are aerosols distributed in the vertical column over the study site at KAUST?
3. What is the seasonal or diurnal variability in the vertical distribution of aerosols?
4. How do prevailing land and sea breezes affect dust emissions and distribution over
   the study site?

This paper is organized as follows. We present a description of datasets and methods in section
two, where we describe the observational datasets used and the WRF-Chem model settings. In
section three, we present the results. More specifically, we explore the first and second research
questions listed above in sections 3.2 and 3.3, respectively. Results presented in sections 3.3 and
3.4 are relevant to the third research question. Section 3.5 addresses the fourth question. We
present a general discussion of the results along with the limitations of our research in section 4.
Finally, we present the key conclusions in section 5.

**2. Data and Methods**

**2.1. Study site**

The KAUST campus is located in the western Arabian Peninsula, on the east coast of the Red
Sea (22.3° N, 39.1° E). This area is affected by local dust storms originating from surrounding
inland deserts, by distantly-generated dust arriving from northeast Africa through the Tokar gap
(see, for example, Kalenderski and Stenchikov 2016; Albugami et al., 2019; Kumar et al., 2019),
and by dust from as far away as the Tigris-Euphrates regions (Parajuli et al., 2019). Therefore,
dust is present in the atmosphere over the study site for the entire year.

Although our focus in this study is on dust aerosols, which are the dominant aerosol over the
study site (Prakash et al., 2014; Parajuli et al., 2019; Ukhov et al., 2020a), some additional
aerosol types also contribute to the aerosol loading at KAUST. Our site is located on the coast;
thus, sea salt aerosol, which is of natural origin, inevitably contributes considerably to the
atmospheric aerosol loading. Furthermore, the study site has several industrial areas nearby that
produce anthropogenic emissions of sulfur dioxide ($SO_2$), and black and organic carbon (BC and
OC) (Ukhov et al., 2020a).

Because the site is located exactly at the land-sea boundary, some unique small-scale processes
exist that affect the local climate of this region. For instance, land and sea breezes affect the

distribution of dust in the atmosphere over the study site. The desert land heats up during the
day, which consequently heats the surface air above the land. This warm air mass rises due to
convection, creating a local pressure 'low' at the surface. The cooler and more moist air over the
Red Sea then flows towards the low pressure zone, thus forming sea breezes (Simpson, 1994;
Miller et al., 2003; Davis et al., 2019). During the night, this flow is reversed to form land
breezes, when the land surface temperature cools quicker than the sea surface temperature.
Because these breezes are driven by the thermal contrast between the land and the sea, their
strengths vary by season. These breezes are further enhanced because of their coupling with
slope winds generated on the Sarawat Mountains, which run along the western coast of the entire
Arabian Peninsula (Davis et al., 2019).
Land and sea breezes affect dust aerosol emissions and transport in our study region. When the
land and sea breezes are strong, they can cause dust emission from the coastal regions. Although
breezes are not responsible for long-range transport, they can affect the local distribution of dust
aerosols over the study site.
Because dust/aerosols are present over the study site for most of the year, they can also interact
with the meteorology and thus affect atmospheric winds and temperature at different time scales
(Jacobson et al., 2006; Rémy et al., 2015). Land or sea breezes are strongly coupled with
dust/aerosols and temperature variability, especially near the surface (Crouvi et al., 2017).
**2.2. Observations**
We use several datasets to evaluate our model simulations and derive the average season profiles
of aerosol loading and surface winds during 2015-2016, as described below.
*2.2.1. Datasets*
We collected meteorological data, including wind speed, temperature, and humidity from a tower
established at KAUST in 2009 in collaboration with WHOI (Woods Hole Oceanographic
Institution) (Farrar et al., 2009; Osipov et al., 2015).
We use cloud-free aerosol extinction profiles retrieved from CALIOP onboard CALIPSO for
analyzing the vertical structure of aerosols at the study site. CALIPSO is flown in a sun-
synchronous polar orbit and is a part of NASA's Afternoon (A-train) constellations (Stephens et
al., 2018). We use level-3 day/night aerosol data v3.00, which are monthly aerosol products
generated by aggregating level-2 monthly statistics at 2° (lat) × 5° (long) resolution (Winker et
al., 2013). The data have 208 vertical levels up to a height of 12 km above sea level.
We also analyze aerosol optical depth (AOD) data from the AERONET station at KAUST
(Holben et al., 1998). We use a level 2.0 data of directly measured AOD values (direct sun
algorithm), which are cloud-screened and quality-assured. From AERONET, we also use an
aerosol number density and a particle size distribution (PSD) obtained by inversion (Dubovik et
al., 2000) to characterize the aerosol particles in the region. We use the AERONET V3, level 2.0
product, which provides volume concentration of aerosols in the atmospheric column in 22 bins
between 0.05 and 15 microns in radius (Dubovik et al., 2000; Parajuli et al., 2019; Ukhov et al.,
2020a).
We use Moderate Resolution Imaging Spectroradiometer (MODIS) level-2 Deep Blue AOD data
(Hsu et al., 2004), which are available daily, for the whole globe, at a resolution of ~ 0.1°× 0.1°.
We use the latest version of the MODIS dataset (collection 6) (Hsu et al., 2013) because of its
extended coverage and improved Deep Blue aerosol retrieval algorithm, compared to its earlier
version (collection 5). We process AOD data of both Terra and Aqua satellites on a daily basis,
and use the average of the two data products for our analysis. From MODIS, we also use the true
color images for a qualitative analysis of a dust event.
We adopt the Modern-Era Retrospective Analysis for Research and Applications version 2
(MERRA-2) data (Rinecker et al., 2011) for comparing the model simulated AOD and dust
concentrations. Aerosol data from the MERRA-2 dataset assimilate several satellite observations,
including MODIS AOD (Gelaro et al., 2017). We specifically use tavg1_2d_aer_Nx and
inst3_3d_aer_Nv products for getting 2-d AOD/Dust Optical Depth (DOD) data and 3-D aerosol
concentrations, respectively. MERRA-2 data consist of 72 vertical model levels between ~0.23
to 79.3 km.
We also employ 555nm column AOD from MISR onboard Terra satellite archived under
collection MIL3DAE_4, which is a daily product available at 0.5x0.5 degree resolution (Diner,
2009). Because MISR has a wider view with nine viewing angles, MISR identifies thin aerosol
layers more accurately and is more sensitive to the shape and size of particles (Kahn et al., 2005).
We also use the RGB composite from SEVIRI (Spinning Enhanced Visible and Infrared Imager)
instrument onboard the geostationary Meteosat satellite, which is a composite prepared from
specific infrared channels that are sensitive to the presence of dust in the atmosphere (Ackerman,
1997; Schepanski et al., 2007). Dust appears 'pink' in these composite images and is thus
distinguishable from clouds, which are usually shown in yellow, red, or green.
*2.2.2. LIDAR data*
Micropulse LIDAR is a fully autonomous active remote-sensing system in which a laser
transmitter emits light vertically upward, and an optical sensor receives the backscattered signals.
The numbers and the detection time of the backscattered photons provide information about the
aerosols and clouds in the atmosphere. We established the LIDAR site on the KAUST campus as
a part of the MPLNET network in 2014. It operates at a wavelength of 532nm. The data from
this LIDAR (hereafter called KAUST–MPL) is the main basis of this paper.
The colocation of the KAUST–MPL and AERONET station provides an opportunity to get a
more comprehensive microphysical picture when the MPL data are combined with AERONET
sun-photometer measurements. We employ GRASP (Generalized Retrieval of Aerosol and
Surface Properties, Dubovik et al., 2011, 2014), which is an open-source inversion code that
combines different types of remote sensing measurements, such as radiometer and LIDAR
observations, to generate fully consistent columnar and vertical aerosol properties (Lopatin et al.,
2013). We take aerosol characteristics from the AERONET retrieval including size distribution,
absorption, scattering optical depth, and refractive index. These parameters serve as inputs to
GRASP, together with MPL data, to generate height-resolved aerosol fields such as aerosol
extinction, absorption, and mixing ratios.

We combine cloud-screened AERONET radiances and LIDAR backscatter signals to retrieve aerosol properties during the daytime. As AOD data are unavailable during the night, for nighttime retrievals, we use a so-called multi-pixel approach, first introduced by Dubovik et al. (2011) and implemented in GRASP. According to this approach, the retrieval is implemented using a group of observations representing different time and location (e.g., several satellite pixels), to retain the variability of the retrieved parameter. For example, in this study, we invert the closest AERONET measurements obtained the day before and the day after, together with the nighttime LIDAR backscatter data, under some constraints on the temporal variability of the columnar parameters (size distribution, complex refractive index, and sphericity fraction) provided by AERONET measurements. In contrast to other more straightforward retrieval approaches used currently, the multi-pixel technique constrains the retrieval without eliminating the variability within the data. The implemented retrieval approach allows us to retain the variability of columnar properties throughout the night. This approach contrasts with the retrieval approach adopted by Benavent-Oltra et al. (2019), which ignores the variability of columnar properties during the night.

The GRASP algorithm relies on an external cloud masking. Overnight lidar retrievals are performed only when cloud-free AERONET sun-photometric observations are available in the preceding evening and following morning. The AERONET cloud-masking algorithm is considered the golden standard, providing very reliable filtering of thick and broken clouds (Holben et al., 1998). In this regard, only clouds that form specifically at night and are undetectable by sun-photometric observations in the evening and morning could influence our retrieved extinction profiles. At the same time, retrieval of these profiles, to a large extent, relies on detailed columnar aerosol properties retrieved before and after nighttime observations. An attempt to retrieve cloudy profiles under the assumption of cloud-free aerosol columnar properties should result in higher fitting errors, and therefore should be easily detectable.

The retrieved aerosol data has 100 levels in the vertical dimension with a resolution of 75m from 505m to 7700m above sea level. The processed LIDAR extinction data has some data gaps because of the quality constraints applied and cloud filtering (Dubovik et al., 2011). To achieve a complete diurnal picture, we also analyze the raw data of the normalized relative backscatter (NRB) from KAUST-MPL, which gives the total backscatter from both aerosols and clouds at a fine 1-min resolution.

**2.3. WRF-Chem model set up**

We use WRF-Chem (v3.8.1) with some recent updates (Ukhov et al., 2020c) for simulating the emission and transport of dust and other aerosols at high resolution at the study site. The innermost domain (d03), which is marked by a red box in Fig. 1, is centered at KAUST and has a fine resolution of 1.33 km, which is required to resolve the essential features of local wind circulation and breezes. The innermost domain is encompassed by a second domain (d02) having a resolution of 4 km that covers the entire Arabian Peninsula. Although the western boundary of domain d03 appears close to that of d02, there are 40 grid cells in between, which is ten times higher than generally recommended, and is sufficient to ensure a smooth transition across the boundaries. While a further westward extension of d02 could be desirable to better resolve the

synoptic weather phenomena across the Red Sea e.g., through the Tokar gap (Kalenderski and
Stenchikov 2016), such phenomena have a minor impact on the diurnal-scale local sea breeze
circulation in our site, which is the focus of our study. To allow full aerosol exchange and cover
all major sources of dust in the region, we nest the two inner domains within a larger domain
(d01) with a 12 km resolution, which covers the entire Middle East and North Africa (MENA)
region shown in Fig. 1. The key physics and chemistry options used in WRF-Chem are presented
in Table 1.

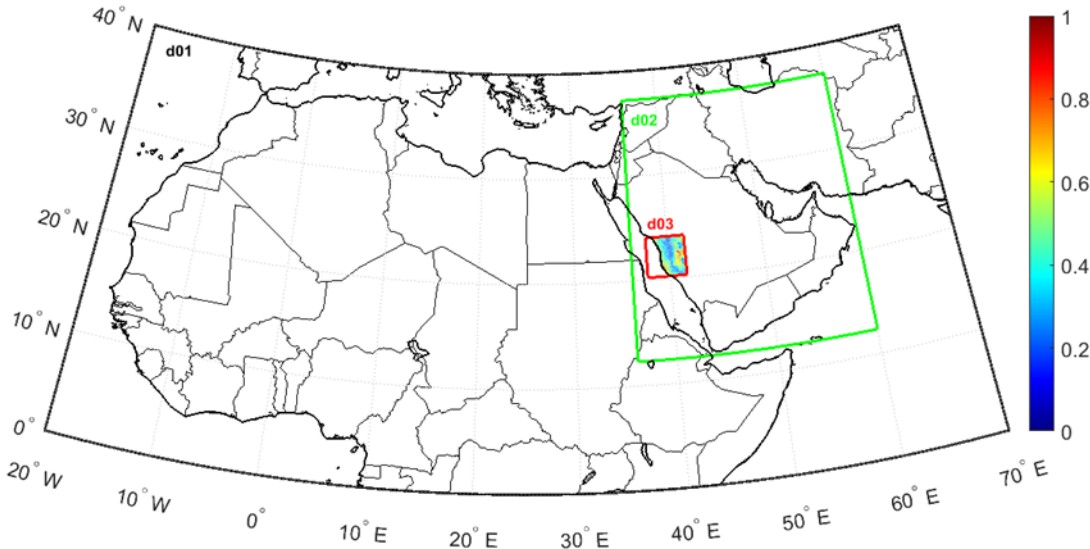


Figure 1. The study region over the Red Sea showing the three nests d01 (black), d02 (green),
and d03 (red) used in WRF-Chem simulations. The base map within d03 shows the high-
resolution dust source function (Parajuli and Zender, 2017) used in this study, in which the
values range from zero to one, with the highest value representing the strongest dust source.
Table 1. Details of key physics and chemistry namelist settings used in WRF-Chem.

| Description | | Namelist Options | References |
|---|---|---|---|
| **Physics** | Microphysics | mp_physics = 2 | Lin et al. scheme |
| | Planetary Boundary Layer (PBL) scheme | bl_pbl_physics = 1 | Yonsei University, YSU (Hong, Noh, and Dudhia, 2006) |
| | Surface layer physics | sf_sfclay_physics = 2 | Monin-Obukhov (Janjic Eta) |
| | Land Surface Model | sf_surface_physics = 2 | Unified Noah land surface model (Chen and Dudhia, 2001) |
| | Cumulus parameterization | cu_physics = 0 (turned off) | |
| | Radiative transfer model | ra_lw_physics = 4, ra_sw_physics = 4 | Rapid radiative transfer model (RRTMG) (Iacono et al., 2008) |
| **Chemistry** | Chemistry option | chem_opt = 301 | GOCART coupled with RACM-KPP |
| | Dust scheme | dust_opt = 3 | GOCART with AFWA changes (LeGrand et al., 2019) |
| | Photolysis scheme | phot_opt = 2 | Wild et al., 2000 |

The model top is set at 100 hPa and has 30 vertical levels between ~20 m to 16 km. To represent
winds better, we apply 'grid nudging' on the u (zonal velocity) and v (meridional velocity)
components of wind above the planetary boundary layer (PBL) in all three domains (Parajuli et
al., 2019). We do not use any convective parameterization scheme and resolve deep convection
in the innermost domain. We employ two-way nesting, which means that the parent domain
provides boundary conditions for the nest, and the nest provides feedback to the parent domain.
The model time steps are set to 72, 24, and 8 seconds for the three domains d01, d02, and d03,
respectively.
Several studies compare the performance of PBL schemes in WRF, showing mixed results under
different model settings (e.g., Saide et al., 2011; Fountoukis et al., 2018; Fekih and Mohamed,
2019). However, these studies have not directly compared the aerosol vertical profiles.
Preliminary results showed that the choice of the PBL parameterization did not have a significant
impact on the vertical distribution of aerosols in our case. In our simulations, we use the YSU
PBL scheme, which is one of the most commonly used schemes, as suggested in the literature
(e.g., Fountoukis et al., 2018; Fekih and Mohamed 2019).
We use high-resolution operational analysis data from ECMWF (~15 km) to provide initial and
boundary conditions in our model, which are updated every 6 hours. The sea surface temperature
(SST) values are also updated in our simulations, using the same ECMWF dataset.
We employ the Global Ozone Chemistry Aerosol Radiation and Transport (GOCART) aerosol
scheme in our simulations (Chin et al., 2002). For calculating dust emissions, we use the AFWA
dust scheme, which follows the original GOCART dust scheme (Ginoux et al., 2001) modified to
account for saltation (Marticorena and Bergametti, 1995; LeGrand et al., 2019). It is important to
represent the dust sources at a fine-scale to capture the smaller-scale physical processes
accurately. Therefore, we use a recently developed high-resolution sediment supply map (SSM)
as the source function (Parajuli and Zender, 2017; Parajuli et al., 2019) in all three model
domains. We adopt the tuning process of the dust model described in Parajuli et al. (2019). We
tuned the model against CALIOP DOD and the same tuning coefficients obtained from Parajuli
et al. (2019) are used in all domains, including the added third domain, which are 0.136, 0.196,
0.120, and 0.110, for DJF, MAM, JJA, and SON, respectively.
We consider dust, sea salt, sulfate, and black and organic carbon (BC and OC) aerosols in our
simulations. Biomass burning and biogenic aerosols are not important over the region, and thus
we do not include them.
Sea salt emissions in WRF-Chem follow the parameterization developed by Monahan et al.
(1986) and Gong (2003). In this parameterization, the rate of sea salt emissions produced via
whitecaps and wave disruption is given as a function of particle size and 10-m wind speed.
We take the anthropogenic emissions of OC and BC from the most recent version of EDGAR
(Emission Database for Global Atmospheric Research) database v4.3.2 available at 0.1°x0.1°
resolution (Crippa et al., 2018). The EDGAR database is a global database that provides gridded
emission maps of several greenhouse gases and air pollutants from 1970-2012. We use OC and
BC emissions data from 2012.
Sulfur dioxide ($SO_2$) is of particular concern because it chemically transforms in the atmosphere
into secondary sulfate, which is an important and influential aerosol at our study site (Ukhov et
al., 2020a, Ukhov et al., 2020b). To achieve a more accurate representation of sulfate aerosols,
we use the $SO_2$ emissions from a time-varying (monthly) inventory developed by NASA for the
same year (2015). This $SO_2$ inventory is developed by combining satellite-based estimates from
the ozone monitoring instrument (OMI) with the ground-based inventory developed by the Task
Force Hemispheric Transport Air Pollution (HTAP) (Janssens-Maenhout et al., 2015), which
provides a more accurate gridded emission dataset with greater spatial and temporal coverage.
The data has global coverage with 0.1x0.1 degree resolution (Liu et al., 2018). This dataset does
not account for $SO_2$ emissions produced by ships; therefore, we take ship $SO_2$ emissions from the
EDGAR v4.3.2 dataset. OMI-HTAP emissions in WRF-Chem are satisfactorily reproduced by
the observed $SO_2$ loading in the Middle East region (Ukhov et al., 2020b).
We activate both gas and aerosol chemistry in our simulations (gaschem_onoff = 1,
aerchem_onoff = 1) and apply the aerosol chemistry options in all three domains.
To determine the contribution of each aerosol species on total AOD, we modify the WRF Chem
code, mainly the Fortran subroutines in *optical_driver.F* and *chem_driver.F* located under the
chem folder. For this purpose, we calculate aerosol optical properties twice, first with the
mixture containing all aerosols and second after removing a specific aerosol. This calculation is
implemented in the subroutine "*optical_averaging*". Thus, we obtain the contribution of specific
aerosol species on total AOD by subtracting the AOD obtained without a specific aerosol from
the total AOD calculated when all aerosols are accounted for.
We calculate the total aerosol concentration (TAC) in $\mu g\ m^{-3}$ by summing up the individual
concentrations of all aerosol species. The equation used to calculate the total aerosol
concentration from the standard output variables of WRF Chem is presented below.
TAC ($\mu g\ m^{-3}$) = [(DUST_1+DUST_2+DUST_3+DUST_4+DUST_5) +
(SEAS_1+SEAS_2+SEAS_3+SEAS_4) + (OC1+OC2) + (BC1+BC2) + P10 + P25] × 1/ALT +
sulf × 1/ALT × 1000 × 96/29.
where, DUST_1…DUST_5 are the dust mass mixing ratios ($\mu g\ kg^{-1}$) in five different size bins;
SEAS_1….SEAS_4 are the sea salt mass mixing ratios ($\mu g\ kg^{-1}$) in four different size bins; P10
and P25 are other anthropogenic PM10 and PM2.5 mass mixing ratios ($\mu g\ kg^{-1}$), respectively;
OC1 and BC1 are mass mixing ratios ($\mu g\ kg^{-1}$) of hydrophobic organic carbon and black
carbon, respectively; OC2 and BC2 are mass mixing ratios ($\mu g\ kg^{-1}$) of hydrophilic organic
carbon and black carbon, respectively; sulf is the $SO_4$ volume mixing ratio (ppmv), ALT is the
inverse of air density ($m^3\ kg^{-1}$), and 96/29 is the ratio of the molecular weights ($g\ mol^{-1}$) of
sulfate and air.
We conduct the model simulations for the entire year of 2015 on a monthly basis (for
computational reasons). For each month, the model simulations start a week before the month
begins, and we discard the data from this week as spin-up. We use data for 2015 only for the
comparison of the model results with other datasets. However, we use the entire two years of
data (2015-16) to derive the seasonal profiles. Because we aim to explore the diurnal cycles, we
use hourly model output data for analysis. While comparing point measurements (LIDAR and
meteorology data) with gridded datasets, we use data from one grid cell containing the KAUST
site from all gridded datasets.

## 3. Results

### 3.1. Surface Meteorology


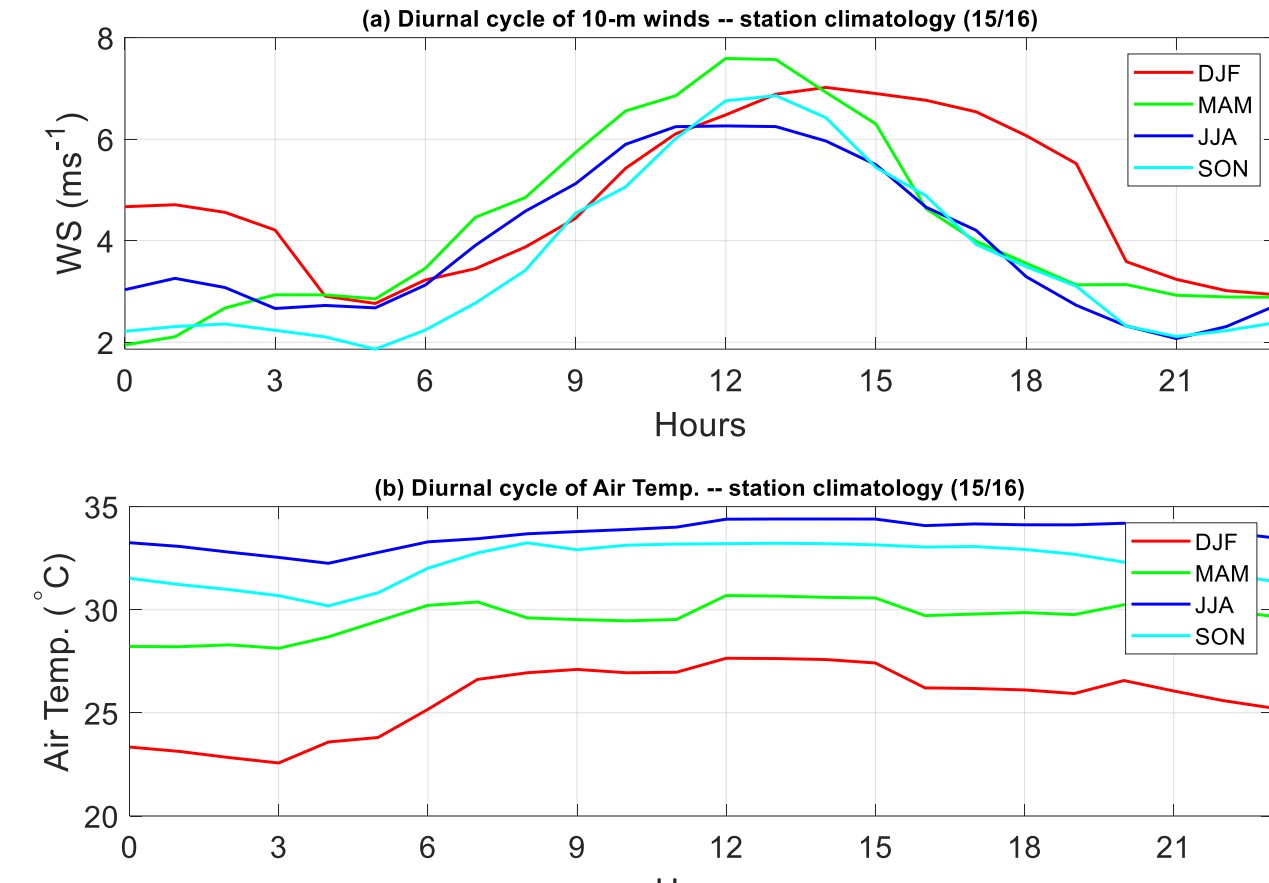


Figure 2. (a) Mean diurnal cycle (2015/16) of surface wind speed (WS) and air temperature
measured at 10-m height at KAUST station. Times are reported in UTC.
Figure 2a shows the mean diurnal cycle (2015/16) of station-measured surface wind speed at the
study site. The surface winds reach a peak around noon UTC (15:00 local time) for all seasons
except winter, consistent with the results of Davis et al. (2019). The aforementioned sea breezes
cause these wind peaks in the afternoon. Note that these sea breezes originate at sea and advance
landward to reach the coast only later in the afternoon (Estoque et al., 1961), where they are
measured at our station. In winter, the wind speed profile shifts to the right, peaking later in the
day at around 14:00 UTC. This shift to later in the day occurs because, in winter, it takes more
time to reach the required thermal contrast between the land and the sea to form sea breezes.
Note the existence of a second peak in the wind speed plot during the night, around 01:00 UTC,
representing the land breezes. These land breezes are stronger in winter than in the other seasons.
The time profiles of air temperature (Fig. 2b) are relatively flat, showing a weak diurnal cycle.
Winter reveals the most pronounced diurnal cycle. The temperature contrast between day and
night is minimal in summer and maximum in winter. The weak diurnal cycle observed in the
station-measured temperature is because of the influence of SST, since the station is located very
close to the sea. The diurnal cycle of land temperature becomes much stronger as we go further
inland in the coastal region (Figure S1), creating a strong temperature gradient between the
ocean and the land surface, which ultimately drives the breeze circulation.
Given the strong diurnal cycles of surface winds and temperature, it is evident that the day and
night circulation in the study area is remarkably different. Therefore, it becomes important to
look at the aerosol vertical profiles separately in the day and night.
**3.2. Model evaluation**
*3.2.1. Surface winds*
Figures 3a and 3b show the diurnal cycle of 10m wind speeds compared with the model
simulations and station data at KAUST for individual months from different seasons chosen to
represent the four seasons. The profiles are in good agreement, although the model slightly
overestimates the wind speed magnitudes. Nevertheless, the model captures the seasonal
variation of wind speed well. These results indicate that our high-resolution simulations
effectively reproduce local features of wind circulations.

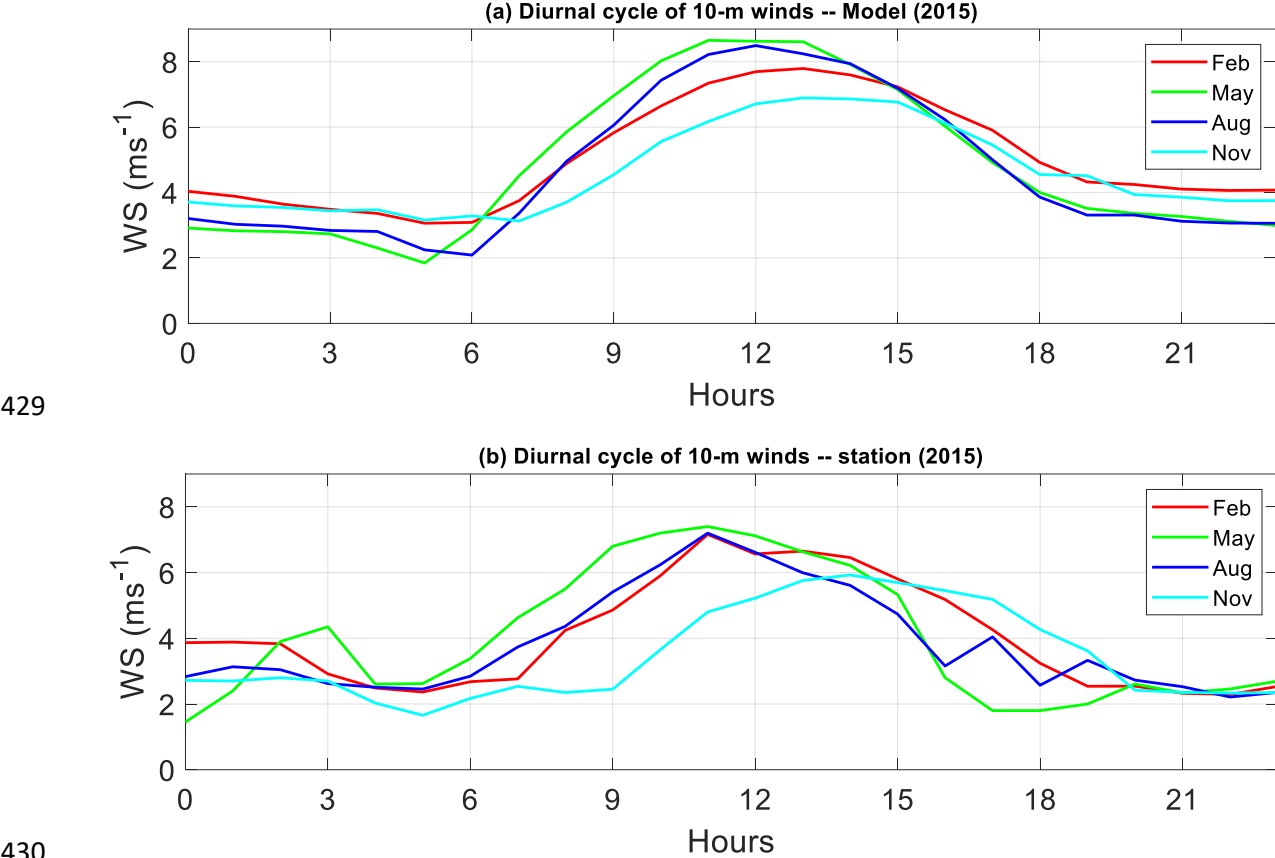



Figure 3. (a) Average diurnal cycle (2015) of 10-m winds at KAUST for four different months
representing each season from (a) model and (b) station. Times are reported in UTC.

### 3.2.2. Comparison of AOD and aerosol volume concentrations

Figure 4 shows the model-simulated time series of total columnar AOD at KAUST obtained
using daily-average values, compared with several datasets, including AERONET, MODIS,
MISR, and MERRA-2. For the model and MERRA-2 data, we only use the daytime data
(between 7 AM and 7 PM local time) to make them consistent with AERONET, MODIS, and
MISR data. In general, all data are consistent and show similar temporal patterns, except during
some large-scale dust events.

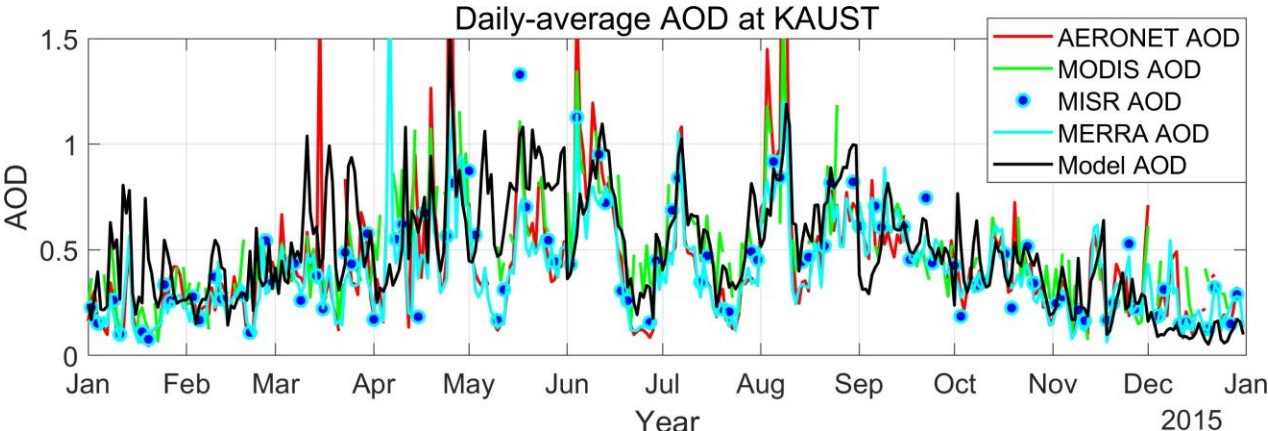

440

Figure 4. Time-series of daily-averaged AOD at KAUST (AERONET, MODIS, MERRA-2, and
Model AODs at 550nm and MISR AOD at 555nm).

For a quantitative evaluation of the model results, we calculate the Mean bias error (MBE) of the
model AOD against the three sets of observations, viz., AERONET, MODIS, and MISR at
KAUST. The MBEs calculated using daily-mean values for 2015 are presented in Table 2. We
also calculate the Pearson's correlation coefficient ($\rho$) of the simulated AOD against the
available observations, after removing the seasonal cycle from all observations. The calculated
MBE for the model is low against all datasets. The MBE is 13.4 % against the most-reliable
AERONET data. The model AOD also shows a good correlation with observations, with a
correlation coefficient close to 0.5 for all datasets. These results demonstrate that the model
simulated AOD values are reasonable.

Table 2. Statistics* of simulated AOD compared with different observations at KAUST.

| Dataset | AERONET | MODIS | MISR |
|---|---|---|---|
| Pearson's correlation coefficient $\rho$ ** | 0.53 | 0.48 | 0.52 |
| Mean Bias Error (MBE) | 0.059 | -0.008 | 0.063 |
| Annual average AOD (Model AOD = 0.49) | 0.44 | 0.47 | 0.43 |

*Calculated using daily-average data for 2015. **all correlation coefficients are significant ($p < 0.0001$).

Figure 5 shows the contribution of different aerosol species on total AOD at KAUST, as
simulated by WRF-Chem. Dust is the major contributor to AOD in all seasons, reaching above
90 % in spring and summer. This result is consistent with earlier reported percentage
contributions of dust over the region (Kalenderski et al., 2016). The anthropogenic contribution
is highest in winter but contributes less than 15 %. The contribution of sea salt emissions is also
small in all seasons (less than 10 %). These results are also qualitatively consistent with the
contributions derived from CALIOP data that use histograms of aerosol type in a grid cell
containing the KAUST site (Fig. S2).

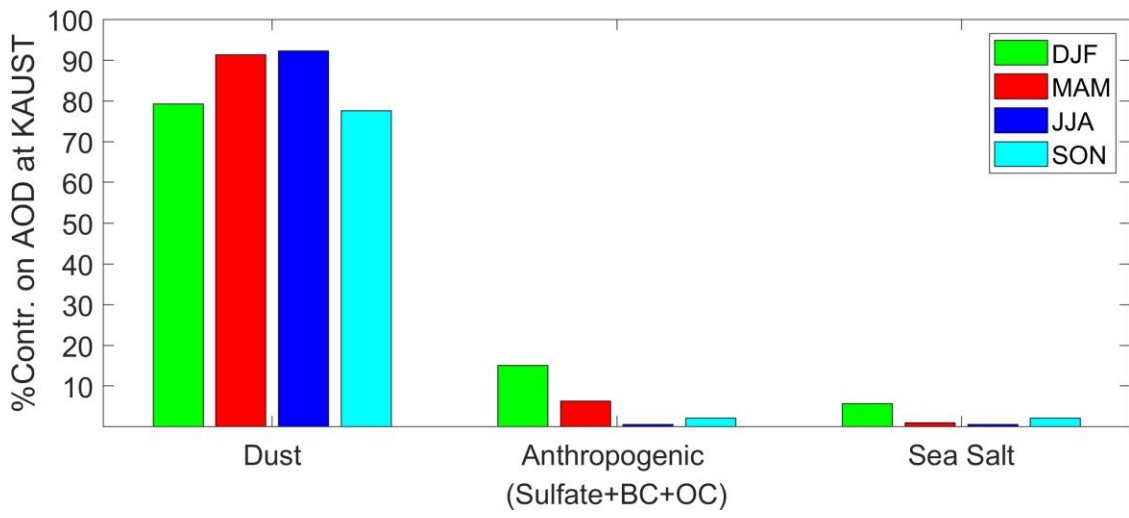


Figure 5. Percentage contribution of different aerosol types on total AOD at KAUST as
simulated by WRF-Chem.
The size distributions of dust, sea salt, and sulfate are modeled in WRF-Chem using
approximations over different size bins. Dust and sea salt are distributed in five and four size
bins, respectively, both between 0.1 and 10 $\mu m$ radius, as detailed in Ukhov et al. (2020a).
Sulfate aerosols are distributed in two lognormal modes, the Aitken and Accumulation modes.
As discussed earlier, dust is the dominant aerosol type; thus, here we compare the volume size
distributions of the modeled dust with the AERONET data. Figure 6 shows the column-
integrated volume PSD in the model and AERONET data. The simulated and observed volume
PSDs are reasonably well matched in all seasons even though the dust in the model is distributed
in five bins only (Parajuli et al., 2019; LeGrand et al., 2019). Although the maximum radius of
particles in AERONET data is 15 microns, which is larger than the maximum size in the model
(10 microns), the majority of particles in the AERONET data fall within the 10-micron range.
Recent measurements from aircraft have shown that dust particles can be much larger (Ryder et
al., 2019), up to 40-micron in radius, during large-scale dust events (Marenco et al., 2018).
However, the optical contribution of such large particles is relatively small. There are two
distinct aerosol modes in AERONET PSD data: one finer mode centered around 0.1 microns,
and another coarse mode centered around 2-3 microns. The coarse mode primarily corresponds
to mineral dust (silt) that originates locally and from inland deserts, northeast Africa, and the
Tigris-Euphrates source region (Kalenderski and Stenchikov et al., 2016; Parajuli et al., 2019).
The composition of the fine mode is much more complex, but usually includes clay particles
transported over long distances and anthropogenic aerosols from pollution sources (mainly as
sulfate) (Chin et al., 2007; Hu et al., 2016; Prospero et al., 1999). The size distributions of sulfate
and sea salt aerosols are presented in the supplementary information (Figs. S3 and S4). Note that
we use the PSD and AOD data from this AERONET station (KAUST) to retrieve the LIDAR
aerosol extinction profiles used in this study.

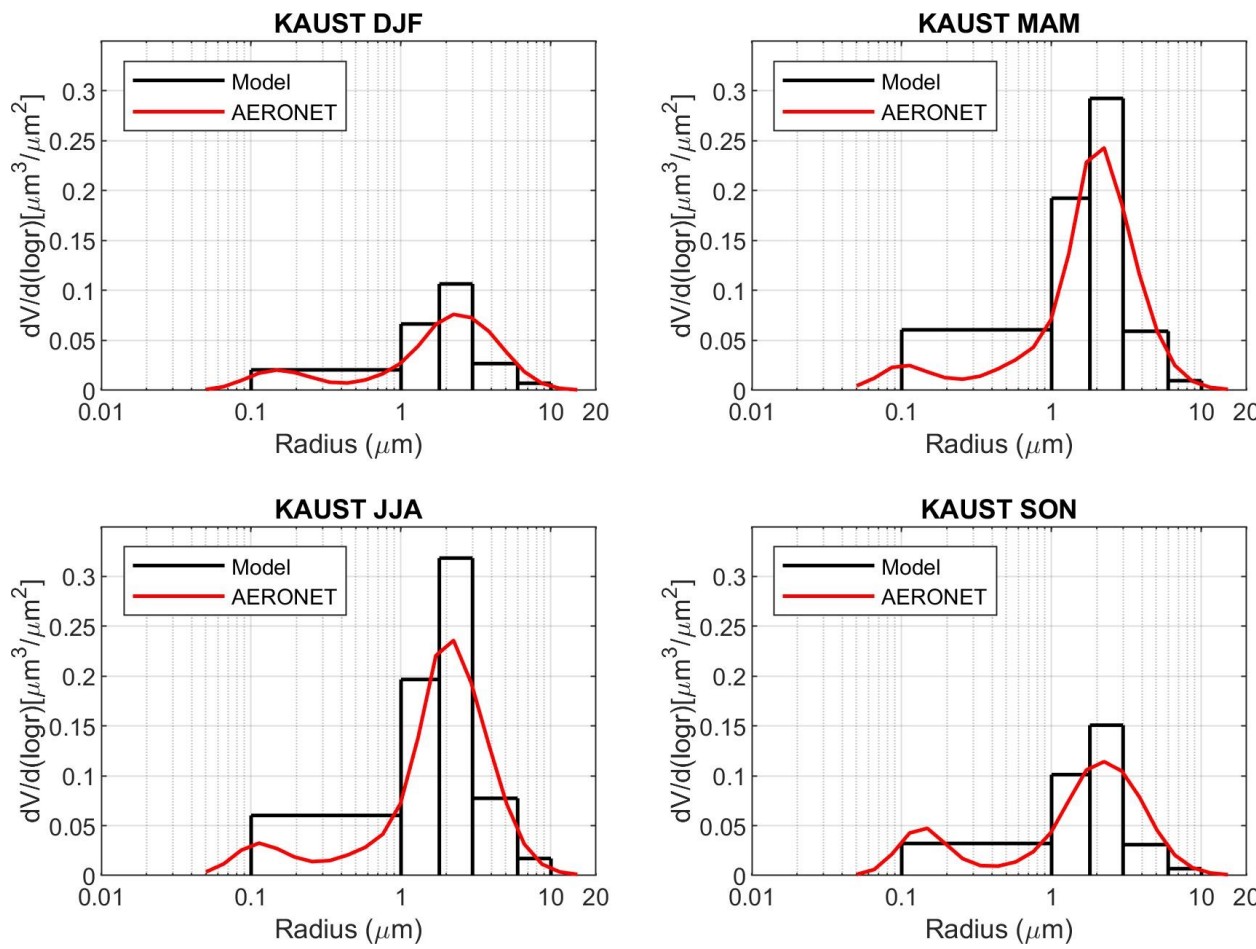


Figure 6. Column-integrated volume size distributions and concentrations of only dust from the
model, plotted against AERONET aerosol volume concentrations at KAUST.

### *3.2.3. Case study of a summer-time dust event*

A large-scale dust storm swept over the KAUST site on August 08, 2015, as seen in the MODIS
image in Fig. 7a. The dust event lasted for two days until August 09. The KAUST AERONET
station registered the second-highest AOD of the entire year on August 08, with the AOD daily
mean reaching 2.48. The AERONET angstrom exponent (AE 440/675) value showed a sharp
reduction on this day, from 0.41 on August 06 to 0.10 on August 08. This reduction indicates the
dominance of coarse-mode dust during the event and that the dust event originated from nearby
inland deserts. By August 09, the dust storm moved towards the south/southwest and spread to a
broader region across the Red Sea and northeast Africa. The MODIS RGB image on August 09
shows a dust plume originating from northeast Africa around Port Sudan, which, after being
deflected by the northerly winds, experiences a marked curvature (Fig. 7b). The SEVIRI RGB
dust composite (Fig. 7c), in which the pink color represents atmospheric dust, also shows strong
dust activity around the KAUST site on August 08.

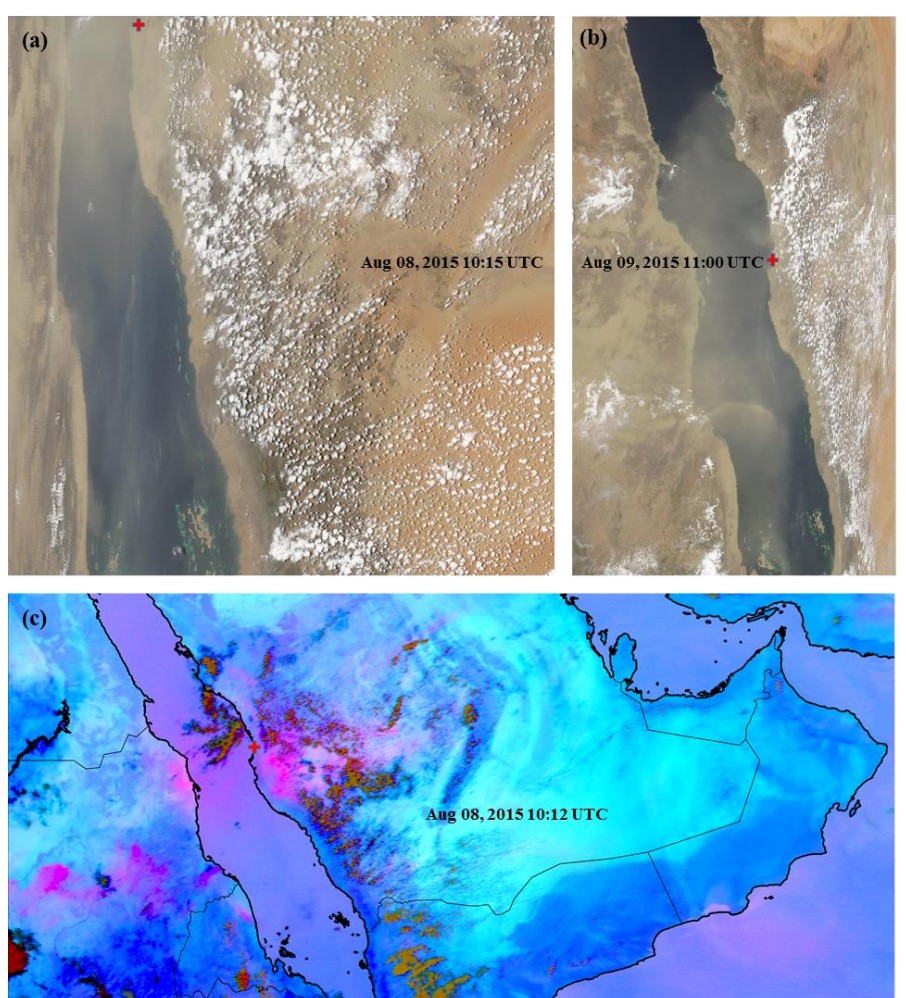

507

Figure 7. MODIS and SEVIRI images during a large-scale dust event. True color images from
MODIS on (a) August 08, 2015 10:15 UTC (b) August 09, 2015 11:00 UTC, and (c) Meteosat
SEVIRI RGB dust composite for Aug 08, 2015 10:12 UTC. KAUST site is marked by a red (+)
mark.

The synoptic conditions of this dust event are somewhat similar to those of a summer-time dust
event reported by Kalenderski and Stenchikov (2016), which was centered over North Sudan.
The dust event we describe here is a typical summer-time dust event caused by high winds
driven by strong pressure gradients (Alharbi et al., 2013). Although haboob-type dust events
commonly occur in the region, analysis of the RGB pink dust composite (Fig. 7c) shows only a
few scattered clouds (red and brown patches) over the study site during this period, ruling out the
possibility of a haboob dust event. Haboob is a typical dust event that commonly occurs in
regions with moist convection, in which dust is generated by strong divergent winds that form
around a cold pool of downdrafts (Anisimov et al., 2018).

521

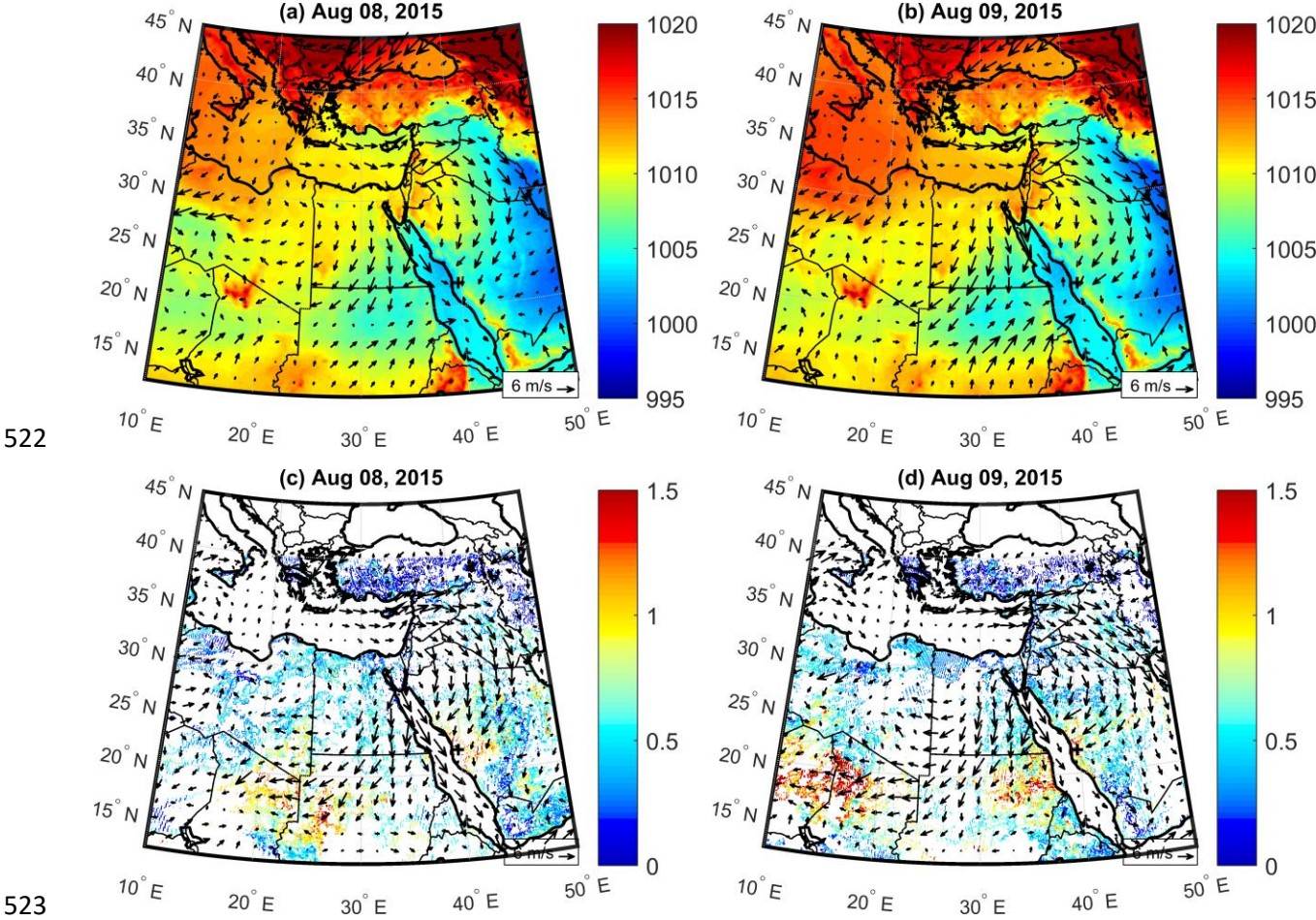

522

523

Figure 8. Mean sea level pressure (MSLP) and wind vectors from ECMWF operational analysis
data during the dust event (a and b), and MODIS deep blue AOD data overlain by model wind
vectors (c and d). KAUST site is marked by a + sign.

As seen in Figs. 8a, b, a high-pressure system developed in the eastern Mediterranean region and
Turkey on August 08, which expanded towards Africa/Middle East and created stronger winds
over the region on August 09. On August 08, a low-pressure system developed, which was
centered on northeast Africa (Sudan). Winds converging towards this low from the
north/northeast adopted a northeasterly flow pattern, which is characteristic of the Harmattan
winds prevalent in the region. The winds originating from the eastern Mediterranean were forced
to curve by the Hijaz mountains in the western Arabian Peninsula, finally converging with the
low-pressure system in northeast Africa and the Red Sea, where the high energy of the flow was
finally dissipated. A high-pressure system persisted throughout the dust event over the Ethiopian
highlands and south Sudan, as shown in Figs. 8a, b. This high-pressure system gave rise to the
southerly/southwesterly winds that also converged towards the low-pressure region around
northeast Africa and the Red Sea.

MODIS AOD also showed a high aerosol loading around KAUST (+ symbol in Figs. 8c, d) on
August 08 that spread across a larger area towards northeast Africa on August 09. Figure 8
shows that the dust mobilization was evidently caused by the northerly/northeasterly winds

moving over the study site. The wind vector patterns are very consistent between ECMWF
operational analysis (Figs. 8a, b) and model simulations (Figs. 8c, d) for most parts of the
domain. This observation is not surprising because we use the ECMWF operational analysis data
for the boundary conditions and apply 'grid nudging' at each model grid using the same
ECMWF dataset. The wind patterns in the two figures differ in some areas, however, especially
over the Ethiopian highlands. Note that the model winds presented are derived from the coarser
12 km domain to show the wind patterns over a larger region beyond our innermost study
domain. In the Ethiopian highlands region, where there is a strong effect from the topography,
such a coarse resolution may not be enough to resolve the fine features of the wind circulations.
At the study site, however, winds are indeed better resolved in our model because the resolution
of the innermost domain is much higher, i.e., 1.33 km.
The model captures the major features of the dust storm reasonably well. Both the model and the
AERONET data register this event as the second-largest dust event of 2015. On August 09, the
model shows a daily average (daytime only) AOD of 1.18 compared to 1.79 given by the
AERONET data (underestimation by ~35 %).
Figure 9 compares the vertical profiles of dust provided by model simulations and the KAUST–
MPL data during the dust event. The right column in the figure shows the simulated dust
extinction coefficient at 550nm, covering the three days during the dust event. Because of the
quality constraints applied in the GRASP algorithm, the processed extinction data from
KAUST–MPL are only partially available during this event. Therefore, we present the raw
normalized relative backscattering (NRB) from the KAUST–MPL to examine the evolution of
this dust event qualitatively, as shown in Fig. 9. Note that around noon local time in summer, the
KAUST–MPL field of view is covered to avoid the sun glare, which is why there is a gap in the
data around this time. In the KAUST–MPL NRB data (Fig. 9, left column), the dust plume
appears as early as Aug 08 (~05:00 UTC) at a height of 1–1.5 km, indicating the onset of the
dust storm. This dust plume becomes strongest by August 09, covering a large part of the
atmospheric column with dust. Although the onset of the dust event is slightly earlier in the
model compared to KAUST–MPL data, the model also shows high dust activity on August 09,
consistent with KAUST–MPL observations. The dust is mainly confined within a height of ~2
km, which is consistent in both datasets.
The model data shows a high aerosol extinction at a height of ~6 km on August 09/10,
particularly at night (Fig. 9), which will be discussed further later. The demise timing of the dust
storm is consistent in both the model and KAUST–MPL data.
When the dust-laden harmattan winds arrive at the Red Sea coast, they encounter the land or sea
breezes depending upon the time of arrival, as discussed further in section 4.3. When they meet
with the opposite sea breeze flow, the air mass rises up, bringing the dust to the upper levels.
Such higher intrusion of dust is evident in the KAUST-MPL data (Figure 9, left) in the
afternoon, during which the sea breezes are most active. The suspended dust is still visible in the
upper levels (~2-3 km) in the night of August 10, because the dust particles have not been
deposited yet.

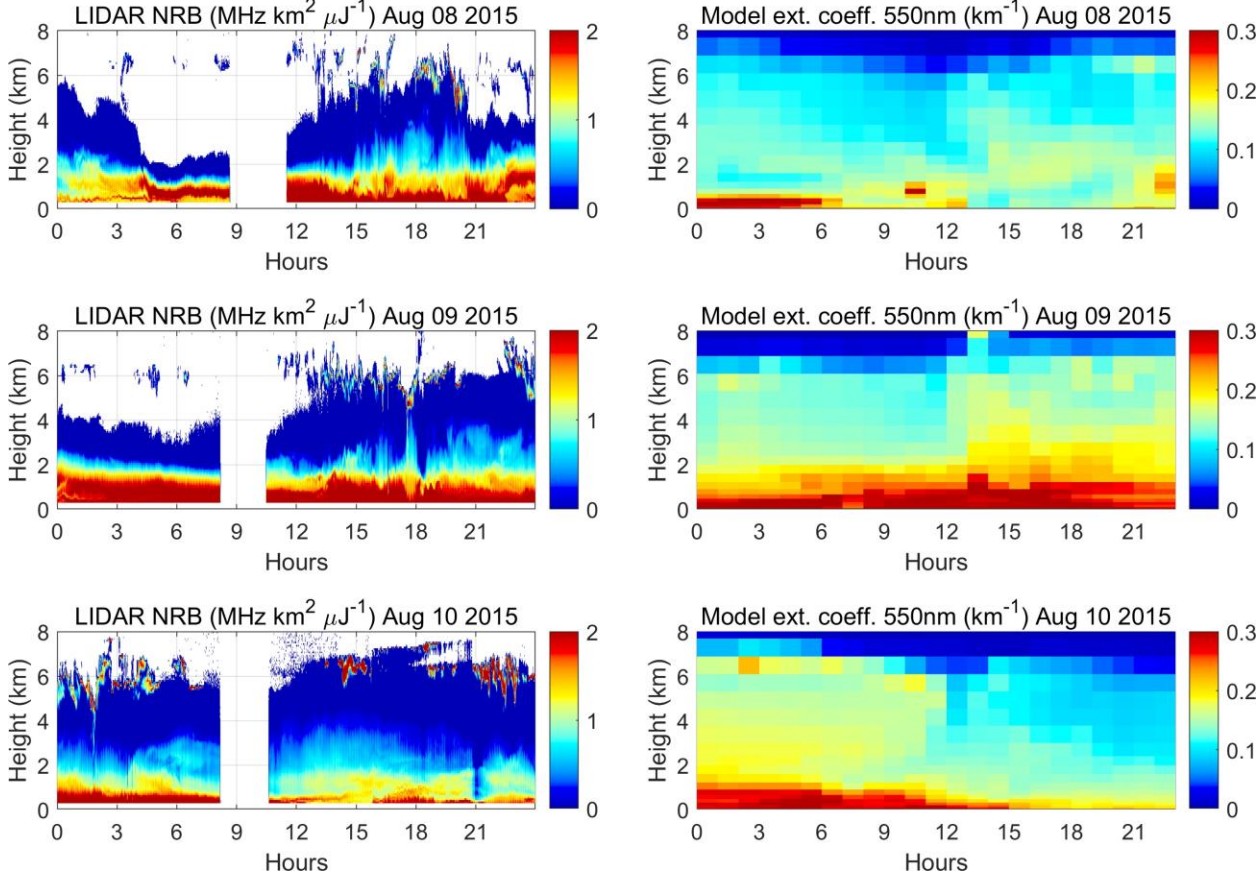

Figure 9. Natural logarithm of normalized relative backscatter (NRB) at 532 nm measured at the
KAUST–MPLNET station (left column) and the model-simulated dust extinction coefficient at
550 nm (right column) during the dust event of August 08/09. Times are reported in UTC.

583

### 3.3. Vertical profiles

#### *3.3.1. Comparison of extinction profiles from KAUST–MPL and CALIOP data*

Figure 10 shows the comparison of aerosol extinction from KAUST–MPL and CALIOP, both of
which show a similar profile. Most aerosols in the atmosphere are confined within the
troposphere below 8 km altitude, which is consistent in both datasets. However, the KAUST–
MPL underestimates the extinctions near the surface compared to CALIOP data. Moreover, the
nighttime dust events observed in the KAUST–MPL data are not present in the CALIOP data.

Day

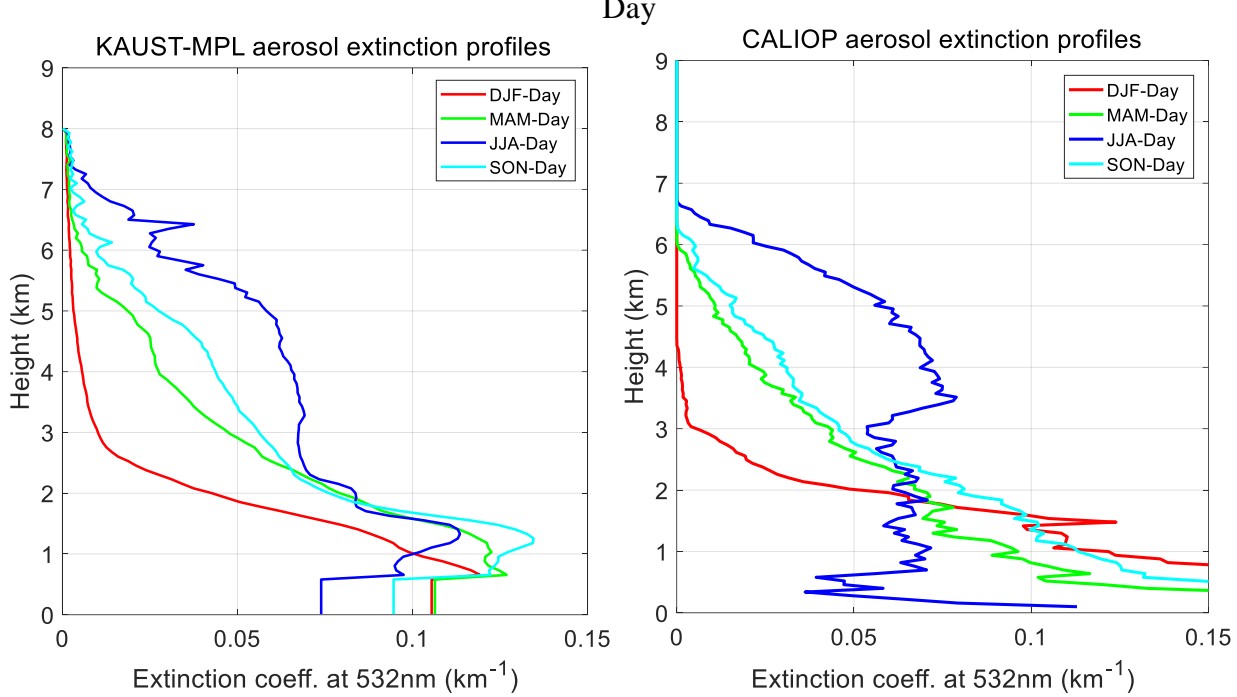



Night

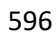

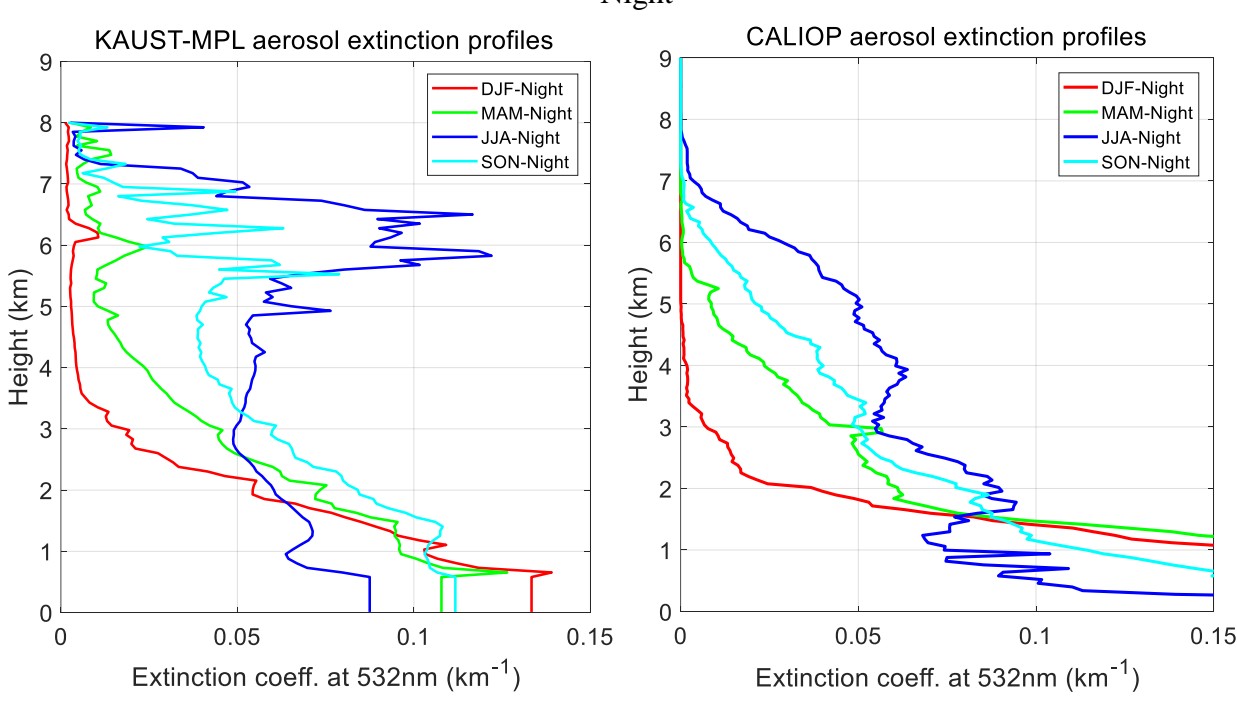


Figure 10. Comparison of seasonal average (2015/16) of aerosol extinction from KAUST–MPL
(left) and CALIOP (right) shown separately for day (upper two panels) and night (bottom two
panels). Heights are above sea level (a.s.l.).

Note that CALIOP extinction profiles represent data averaged over a large grid box (2x5 degree)
that contains the KAUST site. As such, CALIOP represents the larger regional-scale vertical

structure of aerosols compared to KAUST–MPL, which represents a more local structure. Above
~2 km, except for nights during summer and fall, the profiles of the two datasets are much more
similar, indicating the presence of a stable aerosol layer spread throughout the region. This
similarity is understandable because local fluctuations closer to ground level do not penetrate
much above 2 km in winter. Below ~2 km, there are more significant differences between the
profiles. Note that the elevated aerosol loading present in the KAUST–MPL data at about 1-2 km
height is not present in the CALIOP data. It is also worth mentioning that the MPL does not
provide reliable observations in the lowest 550 m, and CALIOP loses accuracy near the surface.
### 3.3.2. Comparison of extinction profiles between KAUST–MPL and model simulations
Figure 11 shows the seasonally averaged vertical profiles of aerosol extinction from KAUST–
MPL and model simulations, shown separately for day and night. The height of the top of the
aerosol layer and the contrast of profiles in different seasons in the KAUST–MPL data and the
model output are similar. The vertical profiles compare reasonably well, with similar orders of
extinction in the daytime, especially considering the range of discrepancy in the KAUST–MPL
and CALIOP data that we discussed above. The magnitude of extinctions in the model and
KAUST–MPL are in good agreement in the nighttime as well, except in summer and spring, in
which cases the KAUST–MPL data shows higher extinctions, particularly above the PBL.
KAUST–MPL data show a distinct aerosol layer located between 5.5 and 7 km, especially in the
nighttime, summer, and the fall. The model does not show such dust layers. KAUST–MPL
daytime data show a typically elevated maximum of dust extinction in the PBL centered around
1.5 km altitude. The model does not identify such a dust loading profile either. The KAUST-
MPL and model profiles agree better in the daytime than in the nighttime, and in winter
compared to other seasons. However, there are no significant differences between daytime and
nighttime profiles in the model. Note that the shape of the profile is reversed during the
nighttime, which the model reproduces weakly. We explore this particularly interesting shape of
the extinction profile at ~1–2 km in the daytime in section 3.4. As discussed later, these unique
features of the profiles are related to the effect of land/sea breezes and topography.

Day

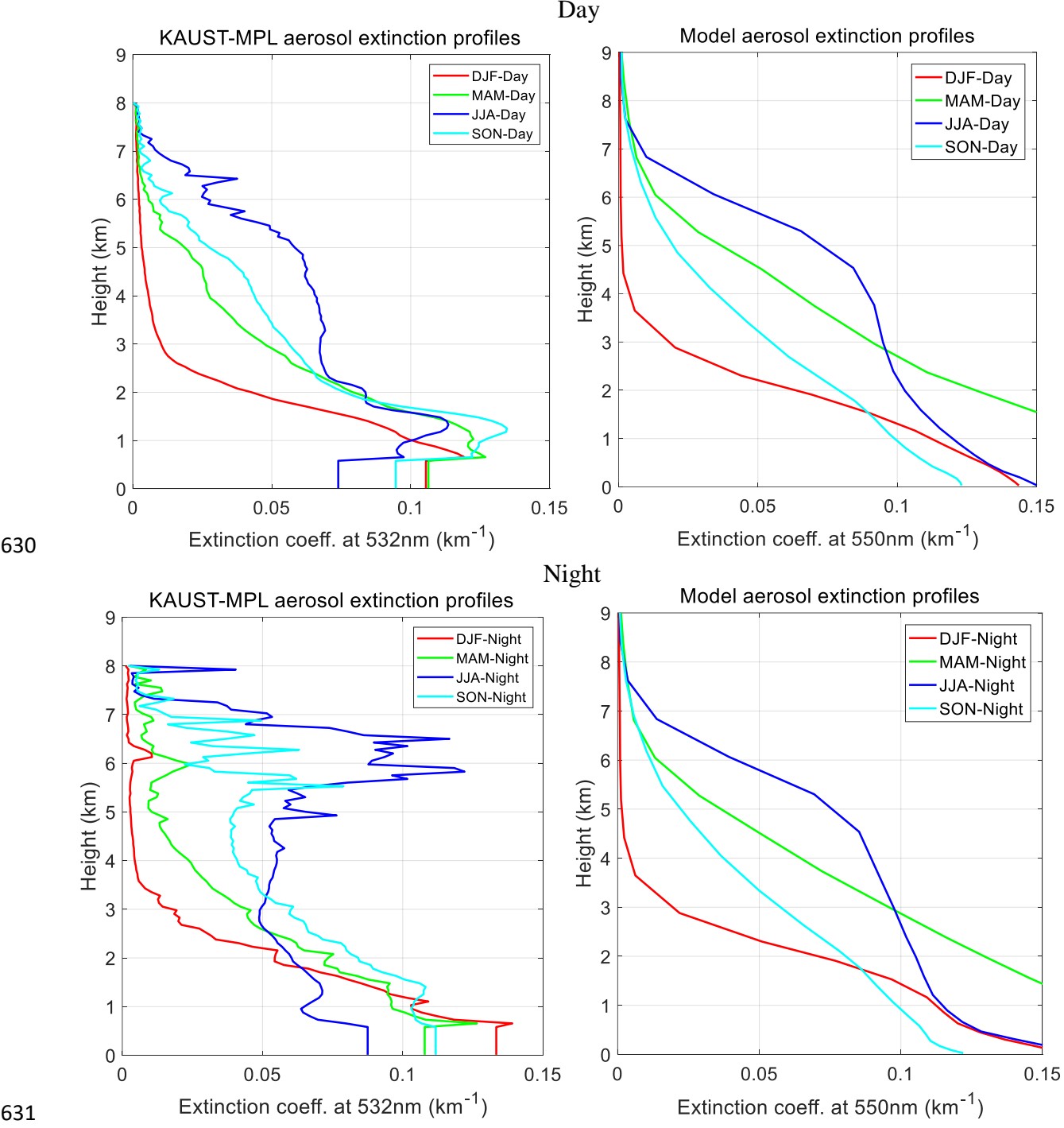

Figure 11. KAUST–MPL retrieved (left column) and model-simulated (right column) aerosol
extinction profiles for different seasons presented separately for the daytime (upper two panels)
and nighttime (bottom two panels). The measurement time of all KAUST–MPL data available
for daytime fall between 05:00 and 15:00 UTC, and nighttime data fall between 17:00 to 02:00
UTC. For the model, the day and nighttime data represent data between these times.
To understand the causes of the elevated dust maxima in the KAUST–MPL profiles at ~1–2 km
altitude in the daytime and 5.5–7 km in the nighttime, we separately analyzed the profiles under
a clear sky and dusty conditions. We define 'clear days' as the days with a daily mean of AOD at
KAUST less than 0.25 and 'dusty days' as the days having daily-mean AOD greater than 0.75,
using either MODIS AOD or AERONET AOD to maximize data availability during large-scale
dust events.
Figure 12 shows the average extinction profiles for clear and dusty conditions from KAUST–
MPL data for 2015/16 obtained using the above criteria. The daytime profile (Fig. 12, left) shows
a similarly elevated dust loading at 1-2 km height, as noted earlier in Figures 10/11, but is much
more prominent. Since 'dusty days' correspond to very high AOD conditions (AOD>0.75)
expected during dust storms, we can infer that the observed elevated dust loading at 1-2 km
corresponds to large-scale dust storms. Studies have shown that this shape is characteristic of
dust profiles observed during large-scale dust events near land-ocean boundaries (Khan et al.,
2015; Senghor et al., 2017). Marenco et al. (2018) also observed a similarly elevated dust
loading over the eastern Atlantic at a comparable height in their airplane observations during the
'heavy dust' period.

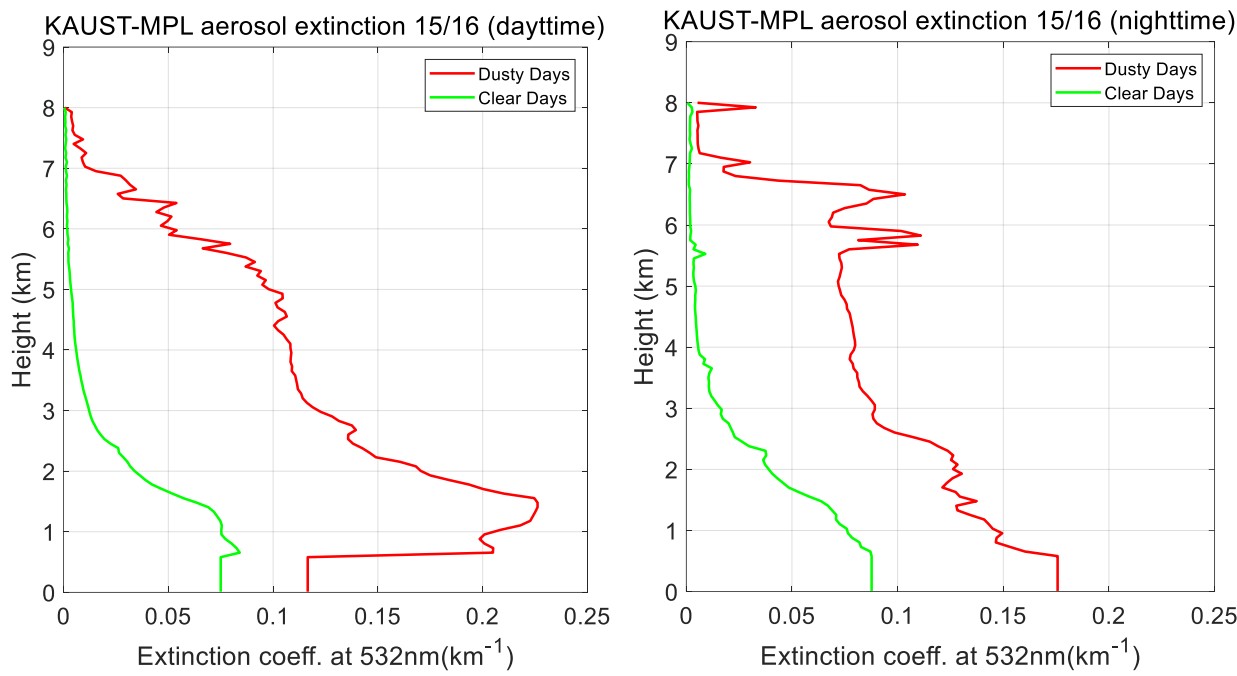

Figure 12. Average vertical profiles of aerosol extinction corresponding to 'clear days' and
'dusty days' from KAUST–MPL data.
The elevated dust layer during the nighttime at the height of 5.5–7 km observed earlier in
summer and fall (Fig. 10/11) is present in the 'dusty days' and is absent in 'clear days' (Fig. 12,
right). The above analysis again tells us that the high dust loadings at 5.5-7 km in the night are
also associated with large-scale dust events. However, it becomes vital to understand the source
of these large-scale, nighttime dust events. Based on our results, we suggest that this nighttime
dust represents transported dust from inland deserts. More vigorous convection in the inland
desert regions during the daytime carries aerosols to higher altitudes. Over deserts in summer,
convection is most energetic in the afternoon. The planetary boundary layer height (PBLH) can
reach well above 5 km (Fig. S5). By the evening, the dust is mixed thoroughly within the PBL by
the strong convection (Khan et al., 2015). At night, the PBL weakens and breaks the capping
inversion (Fig. S6), which allows the dust-laden layer from the PBL to mix into the free
troposphere and be transported to long distances. As an example, we noted such high intrusion of
dust during the night of August 09 (21:00 and 02:00 UTC) in the LIDAR backscatter data of our
case study (Fig. 9). The dust that lies above the PBL is ultimately carried to our site by the
accelerated easterly geostrophic winds (Almazroui et al., 2018), and arrives at our site during the
night. Therefore, the dust layers at 5-7 km observed in the nighttime likely represent dust of non-
local origin transported from inland deserts at higher altitudes.
The dust transport process to our site is evident if we look at the wind vectors at higher altitudes.
As Fig. S7 shows, the winds are northeasterly below ~6 km, which are the regionally prevalent
'trade winds' commonly called Harmattans. Above ~6 km, the winds are easterly. Thus, these
two wind patterns are responsible for transporting dust from the inland deserts to the study site.
The geostrophic easterly wind transports dust at higher altitudes (6–7 km), and Harmattan
transports dust at lower altitudes (1–2 km), which is why KAUST–MPL data shows elevated
dust loading at these heights. In the winter, such transport of dust from deserts to our site is
impossible because the upper-level winds are westerly (Fig. S8).

### 3.3.3. Comparison of vertical profiles of dust concentrations

Figure 13 shows the vertical profile of aerosol concentrations per seasons simulated by the model
compared with KAUST–MPL data and MERRA-2 reanalysis. We have presented these plots
despite their broad resemblance to extinction profiles presented earlier (Fig. 11) because
'concentrations' are more useful from air quality perspective and MERRA-2 provides mixing
ratios of different aerosols rather than extinctions. The variation in concentration profiles in
different seasons is reasonably consistent in all three datasets. The elevated dust maxima at a
height of ~1.5 km observed in the KAUST–MPL profiles is not present in the model or the
MERRA-2 data.  Both the model and MERRA-2 tend to overestimate aerosol concentrations
compared to KAUST–MPL data in summer and in the lower atmosphere, particularly below 1
km. The model-simulated near-surface concentrations in summer are twice as large as those in
the LIDAR data. This overestimation is counter-intuitive because the model AOD agrees well
with the AERONET AOD (Fig. 4) used to constrain LIDAR aerosol profiles. This discrepancy is
related to the size distribution of particles. For AOD to be consistent in the model and LIDAR
data, the model must overestimate the concentration of coarse particles in the lower atmosphere.
Therefore, we can infer that the model overestimates the concentrations of coarse particles in the
lower atmosphere relative to the observed concentrations, which appears to contradict with the
results of Ryder et al. (2019).
In winter, the boundary layer is shallower. The concentration profile resembles a typical profile
that might be expected in a turbulent boundary layer, in which the concentration rapidly
decreases with height, as observed in the field (e.g., Selezneva, 1966) and wind tunnel
experiments (e.g., Neuman et al., 2009). In summer, the boundary layer is deeper, and the strong
turbulent mixing transports dust higher into the atmosphere; consequently, the concentration
profile is steeper.

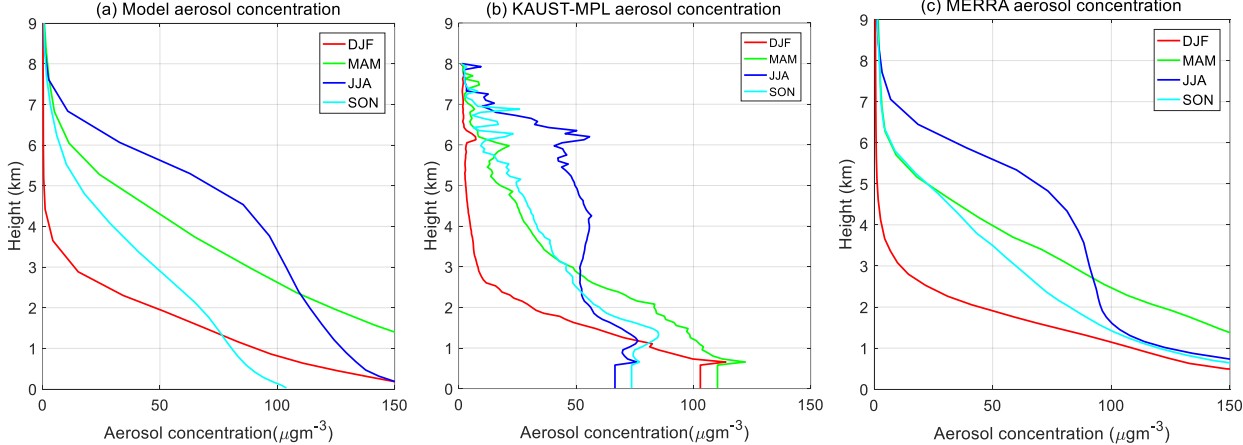


Figure 13. Comparison of the vertical profiles of total aerosol concentrations among (a) the
model (b) KAUST–MPL, and (c) MERRA-2 data for different seasons at KAUST. For MERRA-
2 and model data, total aerosol concentration is the sum of dust, sea salt, sulfate, OC and BC.

## 3.4. Diurnal cycle of aerosols

Figure 14 shows the diurnal cycles of aerosol extinction in KAUST–MPL data across the entire
atmospheric column. Dust is generally confined within the lowest ~2 km in winter and reaches
~6 km in summer, following the seasonal and diurnal variations of PBL. Note that there are some
gaps in the KAUST–MPL data because of the quality controls applied. In summer, there is
significant dust activity in the morning (~ 06:00 local time), and in spring, dust activity peaks
throughout the afternoon. In winter, the KAUST–MPL shows more vigorous dust activity in the
nighttime (21:00 to 00:00 local time) near the surface. This increased dust activity at night is due
to the effect of land breezes, which are strongest in winter (Fig. 2). We explore the effect of
breezes on dust emissions and transport in section 3.5. KAUST–MPL shows high extinctions at a
height of 6–7 km, particularly in the evening of summer and fall, which represent long-range
transported dust during large-scale dust events. Such high-intensity dust events are more frequent
in summer and fall, as observed in the KAUST–MPL data (Fig. 10/11).

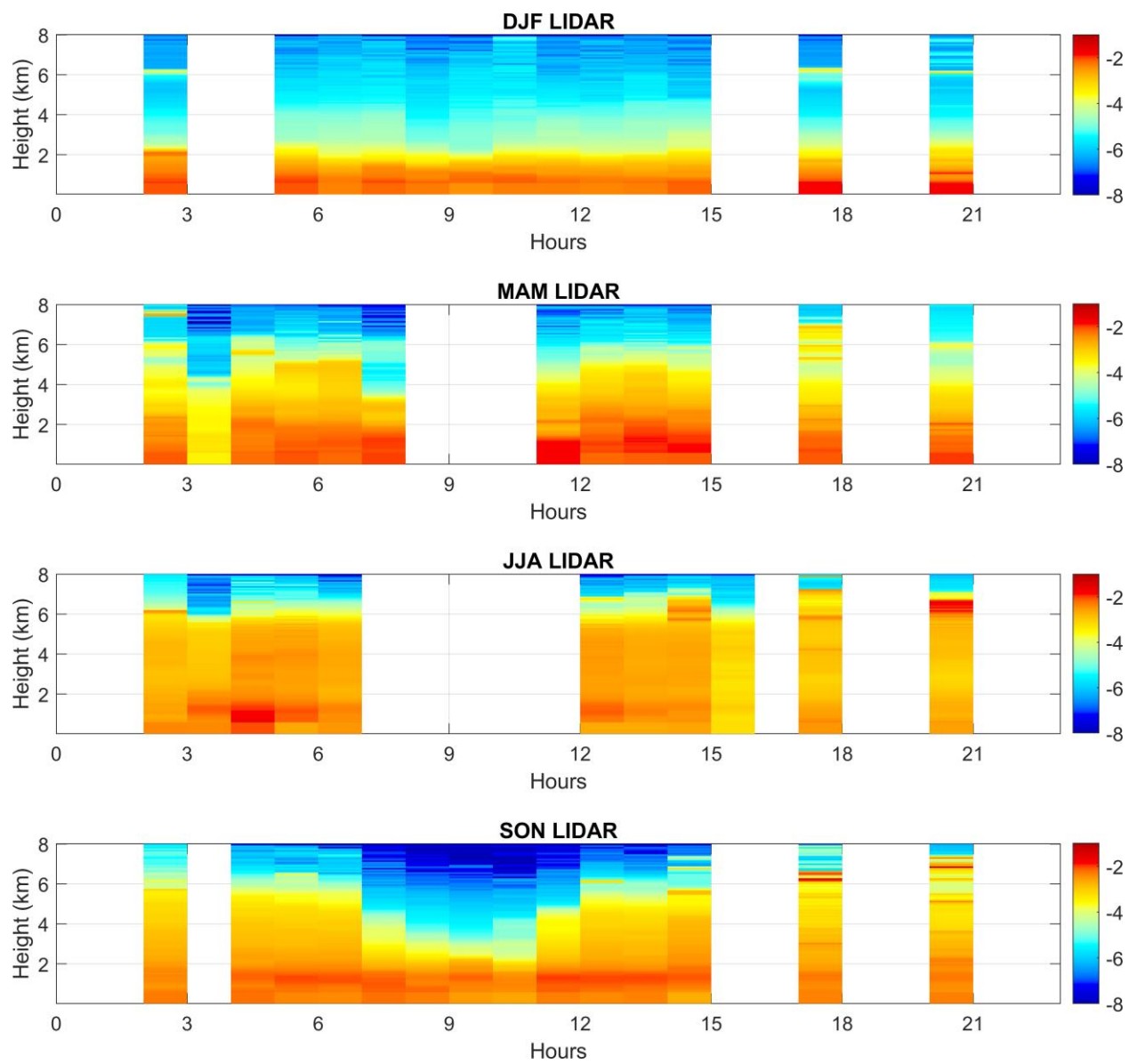

Figure 14. Diurnal profile of the natural logarithm of aerosol extinction coefficient at 532nm $(km^{-1})$ over the atmospheric column observed by the MPL at KAUST. Times are reported in UTC.

## 3.5. Interaction of dust aerosols with Land/sea breezes

Figure 15 shows the circulation features of land and sea breezes in the vicinity of the KAUST–MPL site. The base map in the figure shows the high-resolution dust source function used in this study, where red hotspots represent the most dominant dust sources. Significant dust sources are observed on both sides of the Sarawat mountain range, i.e., the coastal sides and the eastern slopes. Sea breezes are strongest in spring and summer. In contrast, land breezes are strongest in winter and fall. In the daytime, sea breezes penetrate further inland, and the KAUST–MPL site receives northwesterly winds. At night, the KAUST–MPL site experiences northeasterly land breezes, which are strongest in winter.

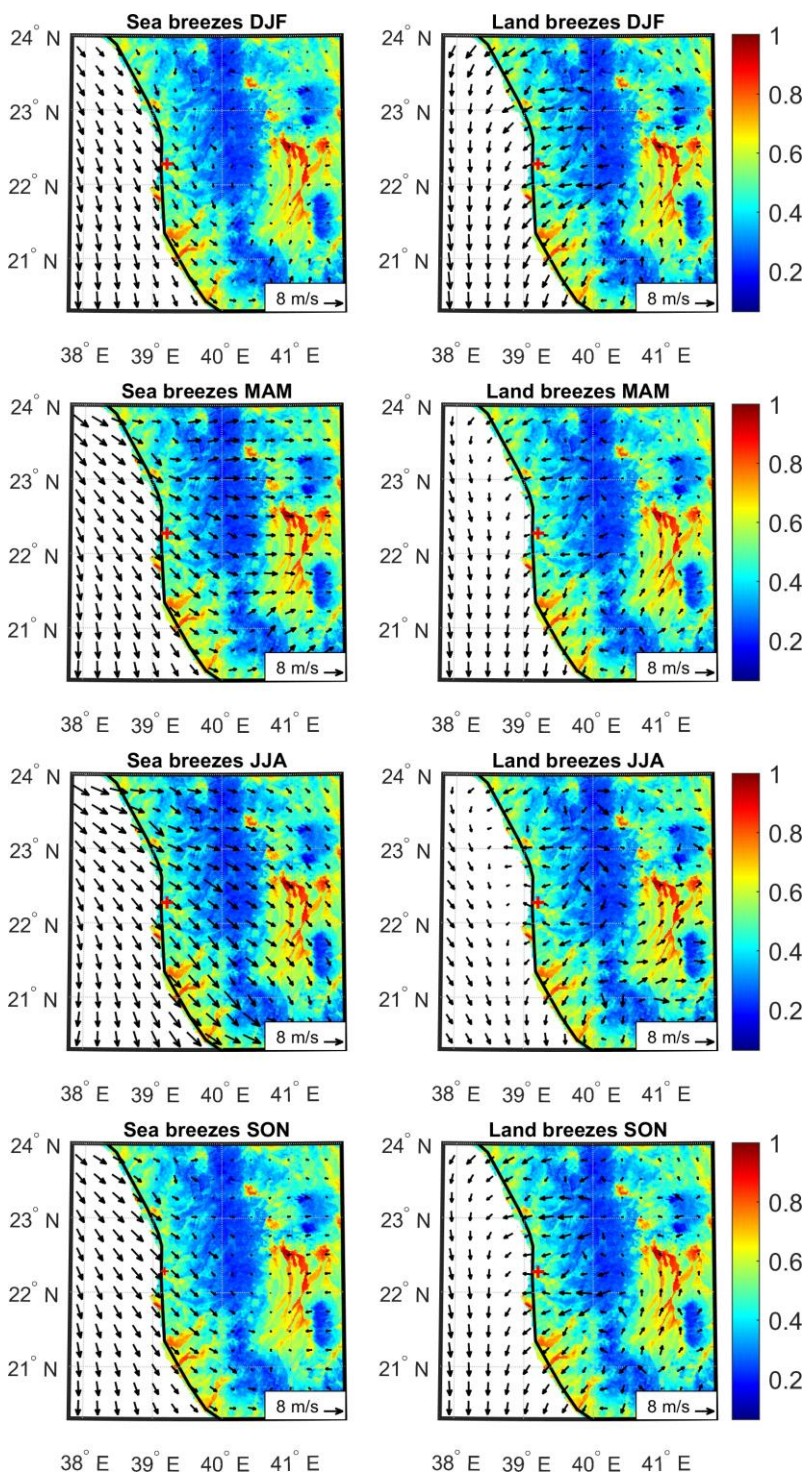

Figure 15. Model 10-m wind speed showing the land (right) and sea (left) breezes. The data are averaged during the peaks of land and sea breezes to highlight their patterns, i.e., 01:00 to 03:00 hours UTC for land breezes (night) and 14:00 to 16:00 hours UTC for sea breezes (day). KAUST site is marked by a red (+) mark. The base map shows the high-resolution dust source function (Parajuli and Zender, 2017) used in this study. The values range from zero to one with the highest value representing the most significant dust source.

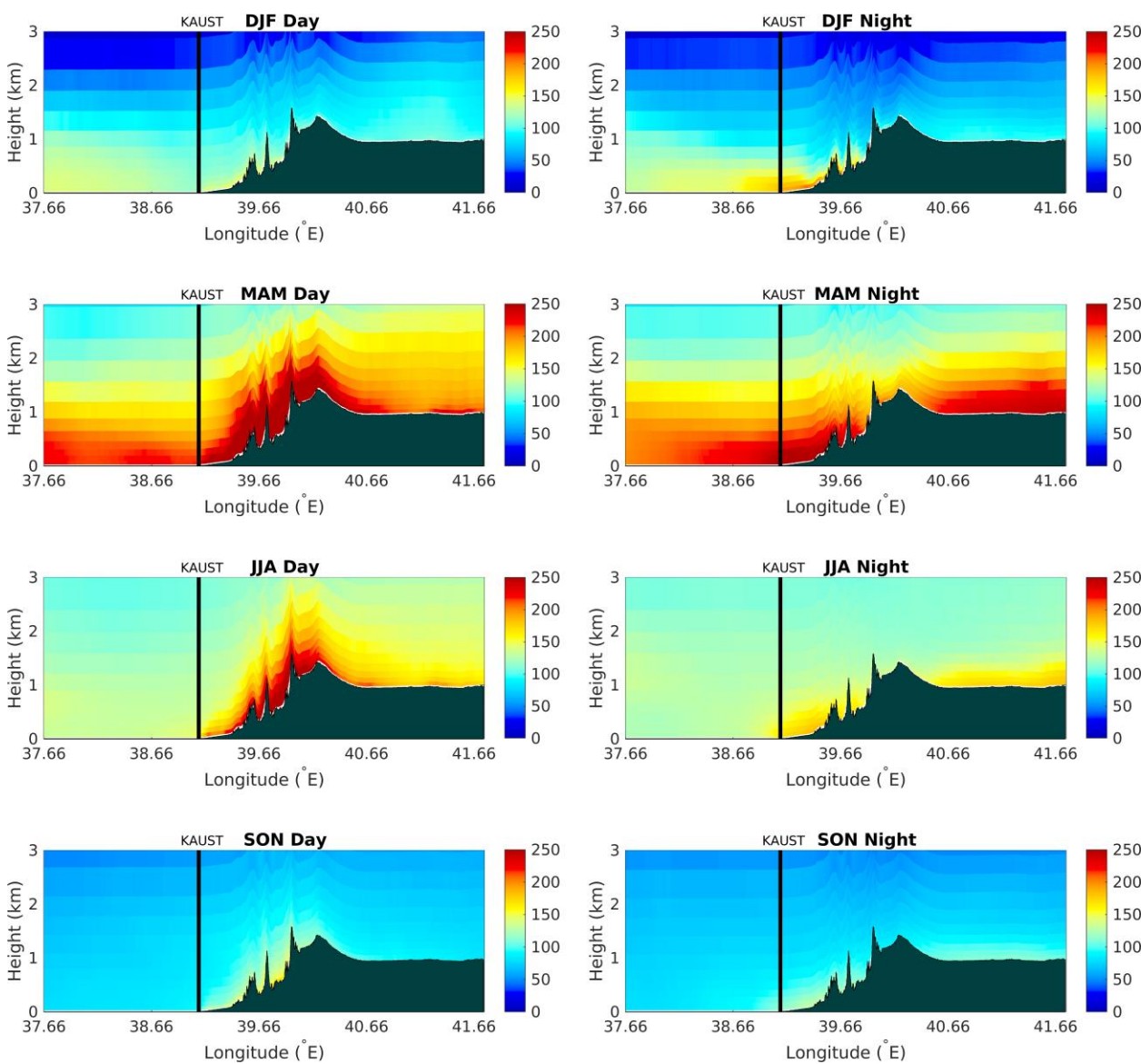

743

Figure 16. Longitudinal cross-section, perpendicular to the coastline, of aerosol concentrations
($\mu g\ m^{-3}$) over KAUST. Data are averaged seasonally and presented separately for the day (left
column) and night (right column). Data averaged during the same period as in Fig. 15 to
demonstrate the effect of land and sea breezes on dust aerosols. The vertical line in black shows
the location of the KAUST site. The land profile along the same section is depicted in black
shades, the top of which shows the actual land elevation.

Figure 16 shows the total aerosol concentration ($\mu g\ m^{-3}$) within the innermost model domain
(d03) in a longitudinal cross-section perpendicular to the coastline over KAUST. The section
also shows the land profile (black shades) where the Sarawat Mountains that run along the
eastern coast of the Red Sea and the relatively flat inland deserts that lie on the eastern side of
the mountains are visible. The mountains reach a maximum elevation of ~1.5 km above sea
level. The effect of land and sea breezes on dust is apparent in Fig. 16, as discussed in further
detail below.

During winter nights, a thin layer of dust collects over the marine boundary layer and the land
near the KAUST site within ~1 km height. This layer is an accumulation of dust that has been
mobilized by land breezes from the coastal plains and the western flanks of the mountains. The
coastal plains of the Red Sea are rich in fine fluvial sediments deposited by wadis, which are
known sources of dust (Anisimov et al., 2017; Parajuli et al., 2019). The western flanks of the
mountains also contain fluvial and intermountain deposits along the slope that are suitable for
resuspension (Parajuli et al., 2014). This mobilized dust is transported towards the Red Sea,
which seems to occur at low altitudes ~500 m (Fig. 16). Some dust collects over the Red Sea
during the daytime in the winter also, which appears well mixed within the relatively shallow
PBL. During the day, the northwesterly sea breezes move landward preventing the dust emitted
from the coastal region from moving over the sea. Therefore, this dust observed during the
daytime must be the residual dust that accumulated overnight. The dust mobilization from the
coastal area by the sea breezes (daytime) is weaker during the winter.
In the spring, there is very high dust loading over the coastal region and the western flanks of the
mountains, which is much higher than in winter. This higher dust loading is consistent with
stronger sea breezes in spring than in winter (Fig. 15). The highest dust loading is observed over
the slopes of the mountains at a height of 1–1.5 km. Recall that the LIDAR data shows a high
dust loading at ~1–1.5 km height at the KAUST site. Two factors appear to contribute to this
high dust loading. First, daytime sea breezes mobilize dust locally from the coastal plains and the
western flanks of the mountains. These sea breezes then push the dust inland and upwards along
the slope of the mountains, up to 3 km height. At the same time, the northeasterly Harmattan
winds also bring dust from the nearby inland deserts towards the mountains. This dust is further
uplifted when the dust-laden Harmattan winds encounter the sea breezes coming from the
opposite direction. Thus, the interaction of sea breezes with the northeasterly Harmattan winds
across the mountains mainly determines the vertical distribution of aerosols over the region. At
night, the sea breezes as well as the PBL weaken, and the vertical extent of dust in the
atmosphere reduces. The land breezes also appear to transport the dust towards the Red Sea from
the western flanks of the mountains at night.
In summer, the patterns of dust mobilization and transport are similar to those in spring but are
not quite as pronounced. In fall, the mobilization of dust from the coast and its ocean-ward
transport is very weak, and the patterns are similar to those in winter.
Figure 17 shows the daytime and nighttime winds at three altitudes for two specific months in
summer (August) and winter (February). Note that the winds are shown at different levels for
August and February to highlight the features of land and sea breezes better. The depth of sea
breezes and land breezes are different, as expected, with the sea breezes being much deeper than
the land breezes, primarily because the PBL is higher during the day than at night. The local
topography also plays a role. Sea breezes are still strong up to a height of ~1150m; however, the
land breezes only reach a height of ~200m. By about 450m, the land breezes subside completely.
The land breeze circulation is confined by the height of the mountains, whereas the sea breeze
circulation extends to a much higher altitude. The returning flow of the sea breezes takes place at
a height of ~2250m in the form of northeasterly trade winds, which are responsible for bringing
the dust to our site from the inland deserts. The return flow of the land breezes occurs at a height
of ~1500m with a change of direction of nearly 180º of the lower part of the subtropical westerly
jets (de Vries et al., 2013) (see supporting information Fig. S6). The variation in the pattern of
these winds along the vertical dimension is generally consistent with the profile of modeled dust
that we presented earlier (Fig. 16).

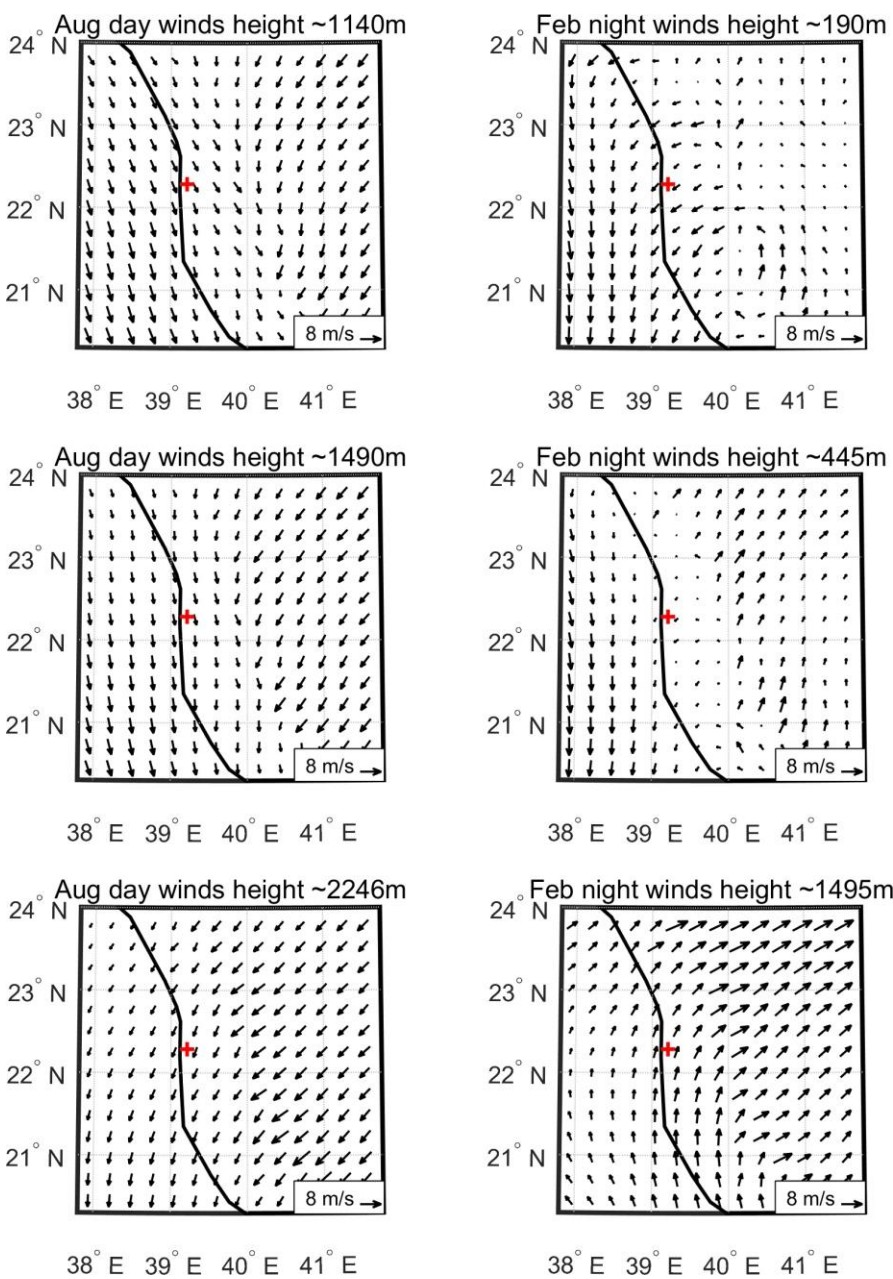


Figure 17. Model (WRF) winds at three different elevations for August (left) and February
(right) within the study domain. The KAUST site is marked by a red (+) symbol.
In summary, the timings and patterns of dust emission and transport in the study region are
evidently affected by land and sea breezes. Note that, across the larger parts of the Arabian
Peninsula, the seasonality of dust mobilization is quite different to our study region, where dust
emission and transport are maximum during summer (Parajuli et al., 2019).
**4. Discussion**
**4.1. Model performance**
The model simulated the surface wind speed at the KAUST site reasonably well as compared to
station data (Fig. 3). Accurately representing the surface winds is vital because the dust emission
is parameterized as a function of friction wind velocity in WRF-Chem (Marticorena and
Bergametti, 1995; LeGrand et al., 2019). Note that dust emissions are generally caused by wind
gusts that occur over very short time scales (seconds) (Engelstaedt and Washington, 2007),
which are much stronger than the average seasonal wind speed displayed in Figure 2. We can
expect these wind gusts to be represented in our simulations because we have used a very small
model time-step (8 sec) in our d03 domain. Given our primary focus is on vertical aerosol
profiles, further analysis of wind gusts is beyond this study's scope.
The model reproduces the AOD time series well in all seasons as compared with several datasets,
including AERONET, MODIS, MISR, and MERRA-2 (Figure 4), with an MBE of 13.4%
against AERONET data. There is some mismatch in the AOD profiles among different datasets
during some large-scale dust events, partly because of the difference in sampling and
measurement frequencies.
The model successfully captured the evolution of a dust event that occurred in 2015 over the
study site in terms of its onset and demise, as well as the height of the dust layer (Fig. 9). Our
results were consistent with several previous studies, such as in Yuan et al., 2019 and Anisimov
et al., 2018. The model generally reproduced the elevated dust layers at ~6 km during the dust
event (Fig. 9), which were prominently seen in KAUST-MPL observations (Figs. 10/11).
However, the model underestimated the AOD at KAUST by about 35 % during the event
compared to AERONET AOD. Simulating these complex, large-scale dust events is extremely
challenging, and thus, we do not expect the model to capture them as precisely, since they occur
only a few times (~2-3) in a year. We note that the performance of WRF-Chem to simulate these
large-scale dust events is case-specific (e.g., Teixeira et al., 2016; Fernández et al., 2019) and
should not be generalized. The model performance was indeed sensitive to the type of dust event
(e.g., Kim et al., 2017), the details of the dust-emission processes (Klose and Shao, 2012; Klose
and Shao, 2013), the dust source function used (Kalenderski and Stenchikov 2016; Parajuli et al.,
2019), and the prescribed size distribution of the emitted dust (Kok et al., 2017; Marenco et al.
840    2018).

**4.2. Aerosol vertical profiles**
The seasonal aerosol vertical profiles were consistent among all datasets that we compared viz.
KAUST–MPL, MERRA-2, and CALIOP (Figs. 10/11/13). These results are consistent with
those reported by Li et al. (2018) over the same region. The WRF-Chem model successfully
reproduced the vertical profiles of dust aerosol extinction and concentration in terms of
seasonality, when compared with the abovementioned datasets. Nearer the surface, the model
showed some disagreement with the observational datasets, as also noted in some previous
studies (e.g., Hu et al., 2016; Wu et al., 2017; Flaounas et al., 2017). Note that such disagreement
between data collected near the surface exists among the observational datasets as well; this
disagreement could arise due to differences in the retrieval algorithms used and the resolution of
the datasets, as discussed in detail below.
The difference in vertical profiles retrieved from KAUST-MPL and CALIOP data could be
related to the differences in the algorithm and resolution between the two datasets. Firstly, while
retrieving aerosol extinction profiles, the CALIOP algorithm uses different prescribed extinction-
to-backscatter (lidar ratio) for a set of aerosol types from a lookup table (Omar et al., 2009;
Winker et al., 2009; Kim et al., 2018). In addition, the CALIOP algorithm has difficulty in
identifying the base of aerosol layers accurately. In particular, the level-3 algorithm ignores the
'clear air' between the surface and the lowest aerosol layer when averaging to avoid
underestimation of extinction in the lower part of the aerosol profile (Winker et al, 2013). In
contrast, the MPL algorithm assumes an averaged lidar ratio for the whole column based on the
aerosol PSD, refractive index, and sphericity, in such a way that it satisfies both AERONET and
MPL co-incident data. Because of the assumption of a constant lidar ratio, MPL retrievals near
the surface could be erroneous, especially when multiple aerosol layers are present (Welton et
al., 2002a). Secondly, KAUST–MPL is a point measurement that captures the temporal evolution
of the dust storms better than CALIOP because it has a higher temporal resolution. For instance,
CALIOP can undersample or overlook some dust events that last only for a few hours. On the
other hand, CALIOP could sample more spatial details of a dust storm because of its extended
coverage along its track compared to KAUST–MPL data. Nonetheless, these two datasets
complement one another, and their combined use can be beneficial in understanding the large-
scale dust storms.
Analysis of the KAUST–MPL data revealed several interesting features of the vertical profile of
aerosols over the study site, which were not documented in earlier studies. For example, we
observed a significant difference between the daytime and nighttime vertical profiles of aerosols.
Some of these detailed features were not apparent in the model simulations. The model
underestimated the nighttime aerosol extinctions at ~6–7 km height in summer and fall compared
to the KAUST–MPL data (Figs. 10/11). Although the model data did not identify these dust
layers at 6–7 km in the seasonally averaged profiles, the model nonetheless correctly identified
these same dust layers during the dust event analyzed in the case study (Fig. 9). This result
supports our speculation that the elevated dust layers at ~6-7 km represent transported dust from
inland deserts during large-scale dust events.
It is difficult to identify the exact reason for the above discrepancy between the model and
KAUST-MPL data, but there are several possible explanations. First, the model could be
deficient in representing the deep convective mixing of dust in the central-peninsula deserts.
Second, although the effect of orography on dust seems to be correctly resolved (Fig. 16), the
long-range transport of dust from the deserts towards the KAUST site may not be fully detected.
Third, part of this discrepancy could also be because of the insufficient model spatial resolution
compared to KAUST–MPL data. KAUST–MPL data is a point measurement while the model
data represents the profiles at a 1.3x1.3 km grid cell, which, although high-resolution, can still
produce a substantial difference, especially in a land-ocean boundary. Finally, the discrepancy
could also be due to the limitation of the GRASP algorithm in handling clouds, because of which
the aerosol layers observed at 5-7 km height in the nighttime could be contaminated with clouds,
as explained further below.
To better understand the origin of two elevated dust layers observed (~1-2 and 6-7 km) and
investigate the possibility of thin-cloud contamination in our MPL retrievals, we analyzed the
volume-depolarization profiles provided by the KAUST-MPL, synchronous to the attenuated
backscatter profiles used in the retrievals. The average volume depolarization value in the lower
atmosphere (1-2 km) was estimated to be 13-14% on average and 7-8% for the upper part (6-7
km) for the selected period. Such values indicate that high extinction values in this altitude range
cannot come exclusively from clouds because pure water clouds generally yield a 1-2%
depolarization value and ~30% or even higher in the case of cirrus clouds (e.g., Del Guasta and
Valar, 2003). The lower depolarization value in the upper part could be explained by the fact that
the aerosol particle sizes are much finer than those in the lower part. At the same time, a lower
depolarization value also suggests the possibility of partial influence by thin clouds. The
presence of thin clouds can probably cause some overestimation of aerosol concentrations and
extinction at these altitudes. However, such an overestimation is expected to increase the fitting
errors, which are easily detectable, as mentioned earlier. To ascertain this with full confidence,
we plan a further analysis utilizing simultaneous retrieval of sun-photometric observations
together with backscatter and volume-depolarization profiles provided by KAUST-MPL in the
future.
Although both model results and the KAUST-MPL retrievals have their own limitations, both
KAUST–MPL and the model data identified two prominent layers of dust over the study site,
one at a lower altitude (~1–2 km) and another at a higher altitude (~6–7 km). These two dust
layers correspond to two different dust sources. The lower dust layer corresponded to dust
originating from nearby deserts, and the upper dust layer corresponded to dust coming from
more remote sources and further inland. The two layers of dust are typical in this region during
dust events. As explained before, a large-scale disturbance usually brings dust from remote
sources at higher altitudes (~6–7 km). When the disturbance comes closer to the site, high
surface winds associated with the disturbance also pick up more dust from nearby deserts giving
rise to a high dust loading at ~1–2 km height. Such stratified aerosol layers have been previously
observed near land-ocean boundaries, where strong temperature inversion occurs, restricting
further mixing of aerosols in the PBL and above (Welton et al., 2002b).
In the lower part (~1-2 km), the atmospheric dust loading is mostly dominated by coarse-mode
particles. In contrast, dust in the upper level (~6-7 km) typically constitutes long-range
transported finer particles. Finer particles can easily reach the upper atmosphere, whereas coarser
particles of higher mass fall back to the surface more quickly due to gravitational settling. Thus,
coarser particles are usually confined to the lower atmosphere, have shorter atmospheric
lifetimes (~1-3 days), and affect hourly/daily scale climate processes such as the diurnal cycle.
On the contrary, smaller particles reach higher altitudes and have longer atmospheric lifetimes.
The extinction cross-section of an individual large particle is bigger than that of a small particle,
but finer particles have stronger radiative effects per unit mass than coarser particles (Khan et al.,
931  2015).

We observed some interannual variability while comparing the vertical profiles for 2015 and
2016, but was not too significant (Fig. S9). Therefore, the observed vertical distribution of dust
aerosols can be considered 'typical' for our region and possibly for other land-ocean boundaries
(e.g., Rasch et al., 2001). This is understandable because the synoptic winds causing large-scale
dust events, and the diurnal-scale breezes that affect the dust distribution, both have strong
seasonality over the study region (Kalenderski and Stenchikov, 2016; Parajuli et al., 2019).
However, as demonstrated by our results, vertical profiles of aerosols can be affected by regional
processes such as breezes, which indicate that the profiles can differ across different regions.
Therefore, it is vital to examine the aerosol vertical profiles of a region to understand the
regional climate.
**4.3. Dust-breeze interactions**

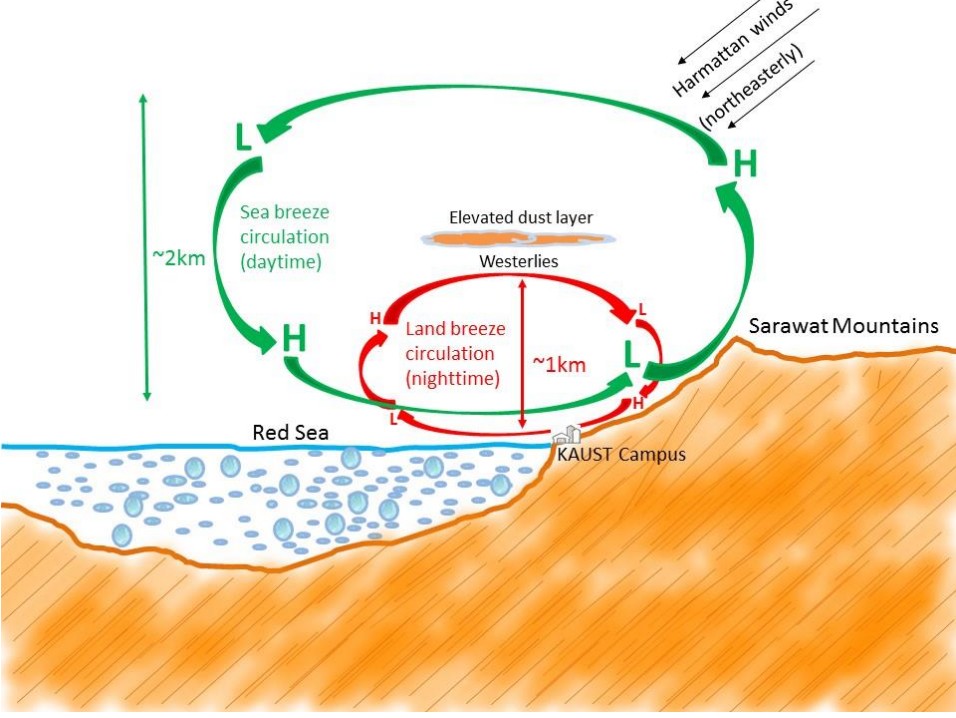


Figure 18. Schematic diagram showing sea breeze (daytime, in green) and land breeze
(nighttime, in red) circulations and dust distribution over the study site at KAUST.
KAUST–MPL-retrieved aerosol vertical profiles also provided an opportunity to understand how
aerosols interact with land and sea breezes over the eastern coast of the Red Sea. The salient
features of the land and sea breezes over the study region revealed by our study are summarized
in Fig. 18. These fine-level interactions are often poorly resolved in coarse-scale simulations.
Our high-resolution simulations (~1.33x1.33 km) nonetheless correctly resolved these features
and showed how breezes affect dust aerosol distribution over the region. Our study is important
because the breezes and dust can directly affect the daily life of populations that reside in the
coastal area. Furthermore, dust over the region affects the surface temperature of the Red Sea
through changes in radiation (Sokolik and Toon, 1998; Osipov et al., 2015, Osipov et al., 2018),
which could have an enormous impact on the Red Sea climate and marine habitats. Additionally,
changes in dust deposition also affect the availability of nutrients delivered to marine ecosystems
(Prakash et al., 2015).
**4.4. Implications of LIDAR data in atmospheric modeling**
MPL data are invaluable for studying the vertical details of aerosols in the atmosphere because
they measure backscatter from aerosols and clouds with a high vertical and temporal resolution
(Welton et al., 2002b; Winker et al., 2009). Most satellite data only provide aerosol properties
over the entire atmospheric column (e.g., Hsu et al., 2004), which are complemented by the MPL
data that provides height, depth, and the particle characteristics of the aerosol layers in the
atmosphere. Since satellite data usually have a low temporal resolution and because many large-
scale dust events are short-lived, MPL data can reveal additional characteristics of dust storms.
In regional and global climate models, it is a usual practice to constrain the total AOD using
some observations (see, for example, Zhao et al., 2010; Parajuli et al., 2019). While such
constraints are desirable because they help to represent columnar atmospheric properties more
precisely, they are not sufficient for certain applications such as air quality modeling, for
example (Ukhov et al., 2020a). Unless the model correctly represents the aerosol vertical
profiles, the model-estimated surface aerosol concentrations may not be reliable. In this context,
KAUST–MPL data can be instrumental in constraining the vertical distribution of aerosols in the
models. Such constraints would ideally benefit the operational forecasting of dust storms and air
quality (Zhang et al., 2015).
Although derived from actual observations, KAUST–MPL retrievals are also subject to
uncertainties, and their accuracy is dependent on assumptions made by the retrieval algorithms.
A study that compared the GRASP retrieval scheme employed here against in situ measurements
showed that the differences were less than 30 % for the different retrieval schemes (Benavent-
Oltra et al., 2019).
**5. Conclusion**
In this study, we investigated the vertical distribution of aerosols over the eastern coast of the
Red Sea. We used data collected from the only operating LIDAR in the region, located on the
KAUST campus, together with other collocated observations and high-resolution WRF-Chem
model simulations, to explore three main aspects of dust aerosols. First, we evaluated how
accurately WRF-Chem reproduces the vertical profiles of aerosols over the study site and
examined its performance during a large-scale dust event of 2015. Second, we investigated the
vertical profile of aerosol extinction and concentrations, as well as their seasonal and diurnal
variability over the study site. Thirdly, we investigated how the prevailing land and sea breezes
affect the distribution of dust over the study site, which is located exactly at the land-ocean
boundary. This study represents a first attempt to understand and describe the interactions
between breezes and dust in this largely understudied region. The main findings of this research
are summarized as follows.
**Model evaluation**
• The simulated AOD obtained from the high-resolution WRF-Chem model setting is
reasonably consistent over the study site across all observational datasets, including
AERONET, MODIS, and MISR. The simulated AOD shows a mean bias error (MBE) of
~13.4 % with the AERONET data.
• WRF-Chem qualitatively captured the evolution of a large-scale summertime dust event
in 2015 over the study site. The model simulated the onset, demise, and the height of the
dust storms reasonably well.
• WRF-Chem simulations show that dust has the highest contribution to total AOD among
all aerosol types, contributing up to 92 % in summer. Anthropogenic (sulfate, OC, and
BC) and sea salt contributions to the total AOD could reach up to 15 % and 6 %,
respectively, especially in winter when both of them are highest.
**Vertical profiles of aerosols**
• Over the study site, most dust is confined in the troposphere, within a height of 8 km. In
winter, dust is confined to lower altitudes than in summer, which is consistent with the
lower PBL height in winter than in summer.
• There is a marked difference in the daytime and nighttime vertical profile of aerosols in
the study site, as shown by the KAUST–MPL data. We observed a prominent dust layer
at ~5–7 km in the nighttime in the KAUST–MPL data. This elevated dust loading is
associated with the dust transported from central-peninsula deserts by the easterly winds
during the night, which is mobilized and lifted up by the preceding daytime convection.
• The seasonally averaged vertical profiles of daytime aerosol extinction are consistent in
the KAUST–MPL, MERRA-2, and CALIOP data in all seasons, which is well
reproduced by our WRF-Chem simulations. The profiles from the different datasets
match better in winter than in summer, consistent with the results of Wu et al. (2017).
**Diurnal cycles**
• There is significant diurnal variation in aerosol loading at the study site in all seasons, as
shown by the KAUST–MPL data. Stronger aerosol activity occurs in the early morning
during the summer, in the afternoon during the spring, and in the night during the winter.
• Both sea and land breezes cause dust emissions from the coastal plains and the western
flanks of the Sarawat Mountains. Such dust emissions are most prevalent in spring.
**Interaction of dust and breezes**
• Sea breezes push the dust mobilized from the coastal plains up along the slope of the
Sarawat Mountains, which subsequently encounters the dust-laden northeasterly trade
winds coming from inland deserts, causing elevated dust maxima at a height of ~1.5 km
above sea level across the mountains.
• The nighttime land breezes are strongest in winter. These easterly/northeasterly land
breezes transport dust aerosols from the coastal plains and the mountain slopes towards
the Red Sea.
• The sea breeze circulation is much deeper (~2 km) than the land breeze circulation (~1
km), as illustrated in Fig. 18.

*Codes and data availability.* The calibrated MPL data used in this study can be obtained from the
MPLNET website https://mplnet.gsfc.nasa.gov/. The source code and additional information
about the GRASP algorithm can be obtained from the grasp-open web site https://www.grasp-
open.com/. MODIS AOD data were downloaded from http://ladsweb.nascom.nasa.gov/data/.
MERRA-2 data were obtained from the NASA Goddard Earth Sciences Data and Information
Services Center (GES DISC) available at https://disc.sci.gsfc.nasa.gov/daac-bin/FTPSubset2.pl.
CALIOP data were retrieved from the website of Atmospheric Science Data Center, NASA
Langley Research Center, available at https://eosweb.larc.nasa.gov/project/calipso/cloud-
free_aerosol_L3_LIDAR_table. ECMWF Operational Analysis data are restricted data, which
were retrieved from http://apps.ecmwf.int/archive-
catalogue/?type=4v&class=od&stream=oper&expver=1 with a membership. EDGAR-4.2 is
available at http://edgar.jrc.ec.europa.eu/overview.php?v=42. OMI-HTAP data are available at
https://avdc.gsfc.nasa.gov/pub/data/project/OMI_HTAP_emis. A copy of the input datasets and
details of the WRF-Chem model configuration can be downloaded from the KAUST repository
http://hdl.handle.net/10754/662750 or by e-mail request to psagar@utexas.edu.
*Acknowledgements.* The research reported in this publication was supported by funding from
King Abdullah University of Science and Technology (KAUST). We thank the KAUST
Supercomputing Laboratory for providing computing resources. We also thank Anatolii
Anisimov for providing SEVIRI images and for helpful discussions. We are grateful to Ellsworth
Judd Welton of NASA Goddard Space Flight Center for the help in archiving and processing the
raw LIDAR data. Thanks are also due to Michael Cusack of KAUST for proofreading the
manuscript.
*Author contributions.* SPP and GLS developed the central scientific concept of the paper. SPP
analyzed the data and wrote the paper with inputs from GLS. IS operated and maintained the
KAUST–MPL site. SPP conducted WRF-Chem simulations, and AU contributed on code
modifications. OD and AL ran the GRASP code. GLS conceived, designed, and oversaw the
study. All authors discussed the results and contributed to the final manuscript.
*Competing interests.* The authors declare that they have no conflict of interest.

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
