# Peer review of "Interaction of Dust Aerosols with Land/Sea Breezes over the Eastern Coast of the Red Sea from LIDAR Data and High-resolution WRF-Chem Simulations"

_Atmospheric Chemistry and Physics, 2020_

## Referee Comment (RC1) · Anonymous Referee #1 · 10 Aug 2020

Review of "Interaction of Dust Aerosols with Land/Sea Breezes over the Eastern Coast of the Red Sea from LIDAR Data and High-resolution WRF-Chem Simulations" by Sagar P. Parajuli et al. submitted to Atmospheric Chemistry and Physics.

This paper is focused on the effect of aerosols onto breeze circulation over the Eastern Coast of the Red Sea employing direct observations and WRF-Chem model. The paper is well structured and written and the results are sound and novel. Specifically, I am very much impressed with WRF-based estimates of the contribution of dust (along with the other components) the aerosol optical depth, the reported consistency

of the vertical distribution of the aerosols across different observational diagnostic and model results and the role of breezes in dust deposition in the coastal environment with complicated orography. I would suggest to single out these conclusions somehow. I suggest acceptance of the paper with few minor caveats and suggestions.

(1) 'Study cite' section (2.1) is rather related to the concept and strategy (BTW given nice fig. 1). It would be useful to rename it accordingly.

(2) Lines 186+ The use of MERRA2 should be better justified. MERRA is not spectral reanalysis and does have some minor to moderate problems over the coastlines and orography. This needs to be commented, better with references evaluating MERRA against alternative products in such conditions. Also in the Conclusions it might be useful to mention as a potential avenue the use of this case study for validating alternative HR products, like ERA5.

(3) ….coordinated in time or/and in space… This requires more elaborate and accurate explanation, otherwise looks very unclear.

(4) Section 2.2 – the arrangement of the domains needs a better explanation, specifically D02 (west boundary). General circulation here is such that requires likely extension of this domain westward. Specifically, there are patterns engaging circulations over the whole western coast (e.g. https://doi.org/10.1175/JHM-D-16-0048.1) which need to be resolved. Try to comment upon potential problems with this. Also the impact of the lateral outer boundary conditions taken from ECMWF analyses should be discussed better (as the choice for the lateral conditions).

(5) lines 345 and around, fig 3. There is an evident seasonal cycle in the AOD distribution – was it removed before computing correlations?

(6) Section 3.2. Diurnal cycle of winds should be better subordinated also with info on wind directions (given the paper focus).

(7) lines 495-497 – analysis of day/night profiles. This para needs edits, as it stands it

is very difficult to handle.

(8) Fig 11 – see comment (6) on wind directions

(9) lines 694+ the interaction of sea breezes with the Harmattan winds is explained in a very wordy and contradictory manner, the para in a whole needs edits.

(10) Fig 15 – change please the arrow scale to see the differences in magnitude in the panels. Also you might wish to use a fine resolution for plotting wind arrows.

(11) Fig 17 and text. What is plotted is MSLP I guess, not surface pressure. Also consider using contours for SLP, as the color is not effective for identifying circulation patterns.

(12) conclusive bullets should be grouped according to the paper flow. Otherwise, they are not convincing, see also general comment.

---

## Referee Comment (RC2) · Anonymous Referee #2 · 20 Aug 2020

This paper presents an approach coupling WRF-Chem, vertical profiles of a MPL lidar and photometric measurements (AERONET) to study aerosols on the western coast of the Arabian Peninsula during 2015. The authors also use MODIS, SEVIRI and CALIOP spaceborne observations to help them in their interpretations. The authors aim to better understand the role of coastal breezes on the vertical distribution of dust aerosols and assess the accuracy of the modelling compared to the observations.

This work is of scientific interest in the sense that the role of breezes and their inter-action with the general circulation of the atmosphere is not necessarily well evaluated

at key locations on the planet, like in the case of the region considered in this article. Dust aerosols are now recognized as having a significant role on the radiative balance of some regions of the globe, but also on economic life (IPCC). This article is therefore interesting, and the results of this research deserve to be published after major revision.

This article should be seriously revised and better organized before publication. There is a lot of repetitions throughout the text, which makes reading the article considerably more cumbersome and detracts from highlighting the main ideas. There is a need to group together the elements of discussion spread throughout the various sections. It is also necessary to be clearer about the objectives as this article can be seen as a publication on the validation of WRF-Chem on the one hand and on the other hand it claims an annual study on aerosols above the experimental site. The part on the cross-comparison between instruments and model should be well separated from the scientific interpretations. A "Model validation" section should be done more directly.

This study is not conducted over a sufficiently long period of time to be able to speak about climatology. It should therefore be repositioned in a more global context to better highlight its scope. A major event has been observed and is the subject of a "case study", but is this event common in other years? Are the observed dust aerosol contents and their vertical distribution throughout 2015 reportable for other years?

The discussion section is confusing and needs to be better organized by a new structure of the article. It would be preferable to separate it from the conclusion, which will then more clearly highlight the major findings of this study.

Other comments

Introduction

L63. The vertical distribution of aerosols has been studied for decades using lidar measurements from the ground-based, aircrafts, and satellites (LITE, CALIOP, GLAS)

platforms. It is indeed an important parameter for the assessment of the climatic impact of aerosols. Numerous publications exist. For deontological reasons, I prefer to let the authors make their complementary bibliography, without influencing them. They can research what has been done during INDOEX, ACE-2 or AMMA at the international level and elsewhere.

L92. Clouds necessarily influence the lidar inversion which usually requires a reference, usually molecular in the upper troposphere. Can you clarify your statement?

Section 2.

Sub-section 2.1. The scheme on the breeze would be better placed in the revised discussion. L157. Use "annual study" rather than "climatology". L163. The CALIOP instrument? L173. Replace version by data? L190. Define DOD. L210 and following. How do you find the absorption coefficient with a MPL? More needs to be said about the implemental retrieval. L240 and following. Aren't there difficulties in parameterizing turbulence at such scales? Can you justify the choice of the PBL scheme? This is an important element for this type of study. L245. Define MENA. L248. Remind the definitions of u and v. L300-306. This approach assumes that there is no internal mixing. L326. Climatology?

Section 3

L343. Give the equation. L350. The "robust" term is somewhat strong with correlations between 0.6 and 0.7. L393-398. Example of duplication. Figure 6a. Define WS. Climatology? L403-406. Already mentioned. L407-417. Combine with what was already mentioned on the sea breeze. L432-433. No, CALIOP inversions use a lookup table with backscatter, color ratio and depolarization as inputs. L440-441. Be careful because the distance between two ground tracks is large. Fig. 7. Height is the altitude a.g.l.? Fig. 7c (MPL during nighttime) and related discussion. What we see above 5 km looks like contamination by semi-transparent clouds (or an average with cloudy profiles). This may also be why there is such a large discrepancy with the model. L458-

460. The difficulty in retrieve aerosols close to the surface is not the same for CALIOP and the MPL. L468. I do not think it is very good in the spring when the model gives much higher values. Sub-section L461. The model does not mark the PBL top well and it gives much higher aerosol extinction coefficients. It would be interesting to see the temporal evolutions of the PBL height deduced from the MPL and the numerical scheme chosen for WRF. A good representation of the PBL is fundamental to take into account the PBL/free troposphere exchanges. Moreover, to compare WRF and the MPL, it would be more interesting to have an OSSE (observing system simulation experiment) as for example in Wang et al. (ACP, 2014). L491-497. I do not understand what is being demonstrated here. Dust aerosol layers are often above the PBL and in coastal areas the PBL is lower. L504-505. Beware of cloud signatures on lidar profiles. L502-503. Aerosols emitted non-locally are most often transported at higher altitudes, above the PBL. This is therefore not an exceptional case. Sub-section L519. I do not understand what the "clear day"/"dusty day" comparison brings to the understanding of the differences between MPL and model. When there is no dust, it is normal that we do not see anything, it doesn't prove anything. L521-522. It is normal that the vertical profiles look like each other as they are proportional, and if the cross-section is not very variable, we find the same vertical structures. L524. As before, the model gives higher values, such as MERRA. The exception is for winter where the agreement is better. L548-550. That is a well-known feature. Sub-section L557. We return to the diurnal cycles as in section 3.2. Figures 6a and 11b show the same information. Why can't we see the same shift over the winter months? Figure 12. With a logarithmic colour scale the contrasts would stand out better. What are the temporal and vertical resolutions? Sub-section 3.4 L581. The effect of the breeze has already been discussed; it should be grouped together. L621-623. So, we don't replicate what the lidar shows. L630-640. What is described here has already been described for different coastal environments, such as during INDOEX. L644-651. There were also significant differences in the profiles in Figure 8, and these should be discussed together. L658. The altitude range of the land breeze is not sufficient to explain the low layer dust aerosols. L660-666.

A typical vertical wind profile would have been interesting. Sub-section 3.5 L711-715. I thought that a haboob was rather generated following the collapse of thunderstorms and the advection of moist air masses. L732. These AODs are much lower than the one announced in L684.

Section 4.

This part is too long. The discussion should be separated from the conclusion. It can also be associated with the analysis of each key element of the article. The organization of the conclusion relating the work presented is confusing.

---

## Author Comment (AC1) · 26 Aug 2020

We thank the reviewer for encouraging feedback on our paper and highlighting our paper's strengths. We agree to reorganize the discussion and conclusion, as suggested. Regarding the proximity of the western boundary of domain d03 with that of d02, we would like to mention that the buffer space between the two borders is more extensive than what is usually recommended. We will discuss more about this in the revised manuscript. We find all other reviews constructive. We will be happy to incorporate them and provide a point-by-point response with the revised manuscript.

---

## Author Comment (AC2) · 26 Aug 2020

We are very much grateful to the reviewer for providing expert insights on our paper. We welcome all the comments and will incorporate them into our revised manuscript. A summary of our response is presented below.

We thank the reviewer for providing us examples of previous studies on vertical aerosol profiles using LIDAR data. We have compiled an extensive review of studies using LIDAR from satellites, field experiments, and networks, as suggested, which will be

[Figure]

included in the revisions. We are far not the first who employ lidar observations for dust profile analysis. However, we would like to highlight that there are no other such studies in the Red Sea coast region. Our research is essential because the presence of breezes over the Red Sea coast affects aerosols' vertical distribution.

We agree that the paper needs better organization, especially the section on conclusion/discussion, which was also pointed by reviewer #1. We also agree that a separate model validation section is preferable. We also recognize that it is more appropriate to use the term 'annual study' than climatology since we only use 2-year data. Regarding cloud contamination in the MPL retrievals, especially at 6-7 km height, it is an essential point because thin clouds can undoubtedly affect the retrievals, and clouds can be confused with aerosols. Even with superior cloud-filtering techniques, we cannot get rid of this problem entirely because, in many cases, clouds and aerosols tend to occur 'together,' especially during large-scale dust events (e.g., during haboobs). Therefore, we agree that some cloud contamination is possible, but we believe that it would not be too much to misidentify a whole aerosol layer. The model does show aerosol layers at that height during the large-scale dust storm (case study). We will be happy to double-check the performance of the GRASP algorithm on cloud screening to ascertain this.

Regarding the PBL height comparison between MPL and model, we have presented the actual model-simulated PBL heights in the supplementary information. In the observations, the PBL height can be taken as the top of the aerosol layers, consistent in MPL and model aerosol profiles (Figure 8).

We will incorporate all other suggestions provided by the reviewer in our revisions. We thank the reviewer again for giving constructive and thoughtful comments.

---

## Referee Comment (RC3) · Anonymous Referee #3 · 1 Sep 2020

General comments on the article.

The overall objective of the paper is to "understand the vertical and diurnal profiles of aerosols over the eastern coast of the Red Sea." This overall aim if the paper is divided into four distinctive questions, the vertical profile, the diurnal and seasonal variation, the ability of WRF-chem to model the aerosols and how the prevailing land sea breezes affect the emissions and distribution of the dust over the study region. I believe this is a valuable scientific study that deserves to be published. The authors have employed appropriate data and analysis to answer the questions. Overall the structure of the

paper need to be re-worked. The authors should consider grouping ideas in the paper in a more consistent manner. The authors need to ensure that the conclusion that they draw are substantiated in the evidence they present. A major short coming of the paper is the attempt to link the dust to the land –sea breeze system – This link is not made successfully. The discussion ignores the fact the there is a massive escarpment in the domain that rises to approximately 1500 m. Acknowledging and accounting for this in itself will not make the link between circulation and dust but cannot be ignored as the Land sea breeze system in this domain is complex and is partly driven by the topography. The link to the dust and the coastal zone completely ignores the fact that the topography will induce its own local and meso-scale wind systems. It is also unclear from the wind data presented when exactly the winds reach sufficiently high speeds to induce these dust storms. The average wind data never exceeds 8 m.s-1. Overall I would focus the paper on the objectives outlined in the paper without trying to link this to land-sea breeze circulations. Detailed comments follow below.

Detailed comments.

Title – I am not sure the title accurately reflects the overall objectives of the paper. The fundamental question posed by the authors is the vertical distribution and the diurnal and seasonal variability of aerosols of the study area. This is as stated by the authors. The land sea breeze is a driver of these two atmospheric aerosol characteristics. The prominence of land sea breezes as expressed in the title is not reflected in the current title of the paper. I suggest the authors re-consider the overall objectives of the paper or modify the title.

Line 36 – "the LIDAR data……remote inland desserts." – The paper provides no evidence that the dust is transported from remote inland dessert sources. In fact the model domain of the dust emissions don't even extend to these areas.

Figure 1 is could be improved by adding a map of the study region. The current figure 1 could be moved to later in the article where the land-sea breeze is discussed which

the authors refer to in line 155.

Line 172 – The author's should add details of the KAUST station. It would be very useful to see the actual location on one of the maps. Also what is the altitude of the station and distance from the coast for example as well as the length of the data series?

Line 216-231 is very difficult to follow. The authors could consider rewording this paragraph to capture the method is a clearer manner. This could be improved by adding more details to the method in this section.

Line 227 constraints should be constrains and "do not" should be "does not"

Line 232-236 – this paragraph is not entirely connected to the previous paragraph and does not stand alone where it is. The authors mention quality constraints applied to the LIDAR data but don't mention what these were or refer to a publication that documents this process.

Figure 2. The colored section of the figure representing the dust source is too small to be useful to the reader at all. If the dust source function is important (which it is) then the authors should add an additional map to show this clearly.

The level of detail in the WRF-Chem model methodology section is not consistent with the detail provided for the other data sets. The authors should consider balancing these sections so that all the study methods are well documented for future studies.

Line 326 – two years of data does not constitute a climatology.

Line 327 and 329 need to be expanded. It is not clear what this means exactly.

Line 337-339 – It is not clear why the authors think the mismatch at this stage is due to sampling and measurement frequencies. The most obvious mismatches are the highest peaks of the AOD values seen in the measurements and not in the model AOD. This explanation premature and not convincing given the model temporal resolution or not accurate or both.

Line 387-391 requires a reference.

Line 393-394 should be re-worded.

Line 396-398 – I am not sure that this sentence does justice to the complex transport associated with this process. Land-sea breezes are local scale wind systems that in the case of this study area could become embedded into to meso-scale winds. The link to long-range transport beyond those scales are complex and associated with multiple embedded systems within regional scale transport. The land-sea breeze mechanism is only a small component of that transport process.

Line 403-404 –the authors need to be specific about what the impacts might be of dust and include references here. Do these impacts have any bearing on the land-sea breeze system directly or on the results of this study?

It would be interesting to see the diurnal temperature pattern of shore of the site. The flat temperature cycle is not ideal for the establishment of a strong land sea breeze system. What creates the temperature gradient shift between daytime and night-time between the land and the sea?

Line 421 – 423- in terms of temperature this is not a justified statement. Even in terms of wind speed data the difference between the day and night values id 6 m.s-1 in MAM while in DJF it is at most 4 m.s-1. These differences may be significant in this region but you need to show that. The figure and text earlier points to a weak diurnal temperature cycle in all seasons.

Line 426-428 –describing all the aerosols as limited to the height of troposphere is not very useful and not a finding that is noteworthy. The vertical profiles of aerosol data in the absence of a vertical temperature profile I believe is difficult to interpret.

Line 455-456 – This needs data or a reference to validate this (or a reference). Also I think you need to refine this discussion as I do not see the same trends as you above 2 km for the two data sets.

Line 471 – 472 - The model does not show this layer in the daytime either. In fact the layer is observed in the nigh-time and not in the daytime in the MPL data.

Line 473-474 – The model daytime and night-time profiles are not very different. I think it is a stretch to infer the model reproduces anything with such a result. The model profile is pretty static for each of the categories graphed . This is over interpretation these data. This should be re-worked.

Line 506-511 – I can't agree with this explanation at all. This requires additional work and temperature profile data to substantiate all the assumptions. The PBL does not break at night and the capping inversion is not broken at night as this is driven in the summer by large scale subsidence which is not dependent on day night changes. The PBL is likely to drop in the evening and possibly a alternative inversion layers form that might trap and concentrate aerosols above. But my explanation is also speculation as it would be easy to see this mechanism from vertical temperature profile data at the very least from the model.

Line 541-Line 550 – This has no context. I don't follow where this has come from in the discussion. Figure 10 - Does not provide a new information about the vertical profile of the aerosols. I am not sure why this discussion could not be combined with the previous section.

Line 568-577 – I am not sure why this was not discussed earlier in the paper in conjunction with figure 6. Also the model and the observations have some real differences in terms of the time of the minimum and maximum values for the different months presented. I think this could be r-worded to more accurately describe what is observed. Again – one year of data is not a climatology!

Figure 13 – I am not sure that the land –sea breeze can be described as covering the entire area of your domain given in Figure 13. Especially if one takes into account that mountainous area lies at about 40 deg E. The wind on the eastern side of the mountainous terrain is almost certainly not associated with land-sea breeze mechanisms anymore but rather on topography induced wind cycles. On the coastal side of the mountains the distance to the coast is about 100 km. Again there has to be a topography component to the wind system in this region which is strengthened by the land sea breeze mechanism.

Line 611-651 – in light of the above I believe this all requires some careful consideration and re-working.

Line 652-666 – this discussion completely ignores the fact that there is an enormous 1.5 km high escarpment sitting in the middle of the domain. This needs to be accounted for in this discussion. The last section of the paper is useful in presenting the occurrence of the high dust events observed in figure 3 that are not captured by the model. This could receive more attention taking into account all the comments above.

The discussion and conclusions need to be revised after the changes are made to the paper.

---

## Author Comment (AC3) · 3 Sep 2020

Thank you very much for providing valuable comments in our manuscript. We find them very helpful, but we want to clarify the queries raised in the review to incorporate the suggestions in the revised manuscript. We agree to structure our manuscript better. The other two reviewers also suggested this. Regarding the link between dust and breezes, we do not consider breezes as a primary generator of dust and a transport system. We agree that the relationships between different aspects of the phenomenon are not evident because the results are scattered in different sections.

Regarding the topographic effect, we extensively discuss them in Section 3.4 (Figure 14), in which the escarpment topography is displayed and discussed along with its impact on dust concentration. Please have a look at this section. The topography indeed has an effect on winds, which we have mentioned in the paper. We understand your concern about looking into more details on surface wind speed, which causes dust emission, and the effect of topography on wind speed. However, our paper's focus is on the vertical profile of aerosols, so we do not go into too much detail on this effect, which can be found in other previous studies (e.g., Davis et al., 2019), as mentioned in our paper.

We also agree to revise the title of the manuscript, as suggested. We will show the diurnal temperature variations at the shore to illustrate the temperature contrast between land and sea, which drives the breeze circulation. We will also revise our text regarding possible mechanisms of dust transport from deserts to our site. There is no simple relationship between temperature and aerosol vertical profile, so the temperature's vertical profile will not really help in our interpretation. We agree that the breezes are formed around the coastal region only, and it is not appropriate to describe them as a control for the whole domain. We will also separate the discussion and conclusion section, which was also suggested by the other two reviewers. We will incorporate all other suggestions into our revisions. We thank you again for your careful review of our manuscript.

---

## Author Response (AR1)

**Dear Editor: 1**

- Thank you for allowing us to revise and improve our manuscript. We are grateful to all the three 2
- 3 reviewers for their expert advice and constructive feedback on our manuscript. We have done our best to
- 4 incorporate all the comments into our manuscript in this revised version. We believe the reviewer's input
- greatly improved the quality of our manuscript. The revised manuscript has also gone through an English 5
- proofreading service. Below is our detailed responses. Reviewers' comments are in black and our 6
- 7 responses are in blue. Texts and quotes from the manuscript are given within quotation marks ("").

**8 Response to reviewer #1**

- 9 Review of "Interaction of Dust Aerosols with Land/Sea Breezes over the Eastern Coast of the Red Sea
- 10 from LIDAR Data and High-resolution WRF-Chem Simulations" by Sagar P. Parajuli et al. submitted to
- Atmospheric Chemistry and Physics. This paper is focused on the effect of aerosols onto breeze 11
- circulation over the Eastern Coast of the Red Sea employing direct observations and WRF-Chem model. 12
- 13 The paper is well structured and written and the results are sound and novel. Specifically, I am very much
- 14 impressed with WRF-based estimates of the contribution of dust (along with the other components) the
- aerosol optical depth, the reported consistency of the vertical distribution of the aerosols across different 15
- observational diagnostic and model results and the role of breezes in dust deposition in the coastal 16
- environment with complicated orography. I would suggest to single out these conclusions somehow. I 17
- 18 suggest acceptance of the paper with few minor caveats and suggestions.
- 19 Thank you very much for pointing out the strengths of the paper. We have revised the conclusions as you have kindly suggested. 20
- (1) 'Study cite' section (2.1) is rather related to the concept and strategy (BTW given 21
- 22 nice fig. 1). It would be useful to rename it accordingly.
- 23 Thank you for the suggestion. Section 2.1 provides a brief description of the study site and introduces the
- 24 breeze circulation, which is typical to the Arabian Red Sea coastal Plain. Breeze circulation is a key
- feature investigated in this study, along with its interactions with dust. Therefore, we would like to retain 25
- the title. However, considering your comment, we have moved Figure 1 to the discussion section and 26
- 27 revised the text with pertaining information about the study site.
- (2) Lines 186+ The use of MERRA2 should be better justified. MERRA is not spectral reanalysis and 28
- does have some minor to moderate problems over the coastlines and orography. This needs to be 29
- commented, better with references evaluating MERRA against alternative products in such conditions. 30 31 Also in the Conclusions it might be useful to mention as a potential avenue the use of this case study for
- 32
- validating alternative HR products, like ERA5.
- 33 We are aware that MERRA-2 might have problems over the coastlines and orography. However, this is
- 34 the only data assimilation product, besides CAMS, which assimilates satellite observations of aerosol
- 35 properties and provides height-resolved aerosol distribution such as aerosol mixing ratios, as it is
- 36 mentioned in the paper. We are aware of the ERA5 data set, which is considered better than its
- predecessor ERA-Interim. However, ERA5 does not provide aerosol concentrations. 37
- 38 (3) . . . . coordinated in time or/and in space. . . This requires more elaborate and accurate explanation, 39 otherwise looks very unclear.
- 40 We rephrased this sentence and further edited the paragraph to make our explanations more clear. The 41 revised paragraph is provided below:

42 "We combine cloud-screened AERONET radiances and LIDAR backscatter signals to retrieve aerosol

43 properties during the daytime. As AOD data are unavailable during the night, for nighttime retrievals, we

44 use a so-called multi-pixel approach, first introduced by Dubovik et al. (2011) and implemented in

45 GRASP. According to this approach, the retrieval is implemented using a group of observations 46 representing different time and location (e.g., several satellite pixels), to retain the variability of th

representing different time and location (e.g., several satellite pixels), to retain the variability of the
 retrieved parameter. For example, in this study, we invert the closest AERONET measurements obtained

the day before and the day after, together with the nighttime LIDAR backscatter data, under some

49 constraints on the temporal variability of the columnar parameters (size distribution, complex refractive

50 index, and sphericity fraction) provided by AERONET measurements. In contrast to other more

51 straightforward retrieval approaches used currently, the multi-pixel technique constrains the retrieval

52 without eliminating the variability within the data. The implemented retrieval approach allows us to retain

the variability of columnar properties throughout the night. This approach contrasts with the retrieval

approach adopted by Benavent-Oltra et al. (2019), which ignores the variability of columnar propertiesduring the night."

56 (4) Section 2.2 – the arrangement of the domains needs a better explanation, specifically D02 (west

57 boundary). General circulation here is such that requires likely extension of this domain westward.

58 Specifically, there are patterns engaging circulations over the whole western coast (e.g.

59 https://doi.org/10.1175/JHM-D-16-0048.1) which need to be resolved. Try to comment upon potential

problems with this. Also the impact of the lateral outer boundary conditions taken from ECMWF analysesshould be discussed better (as the choice for the lateral conditions).

62 We are aware that some large-scale dust storms also take place across Red Sea a few times a year, for

63 example through Tokar gap. We have already mentioned this in section 2.1 with relevant references.

64 Considering the suggestion, we have added the following lines in section 2.2 to clarify this further.

65 "Although the western boundary of domain d03 appears close to that of d02, there are 40 grid cells in

between, which is ten times higher than generally recommended, and is sufficient to ensure a smoothtransition across the boundaries. While a further westward extension of d02 could be desirable to better

resolve the synoptic weather phenomena across the Red Sea e.g., through the Tokar gap (Kalenderski and

68 Tesoive the synoptic weather phenomena across the Ked Sea e.g., through the Tokar gap (Kalenderski and69 Stenchikov 2016), such phenomena have a minor impact on the diurnal-scale local sea breeze circulation

70 in our site, which is the focus of our study."

The ECMWF operational analysis (restricted data used to build initial and boundary conditions for our simulations) is one of the most reliable and high-resolution (~15km) dataset currently available, so the

simulations) is one of the most reliable and high-resolution (~15km) da
 potential impacts from boundary conditions should not be a problem.

(5) lines 345 and around, fig 3. There is an evident seasonal cycle in the AOD distribution – was it
 removed before computing correlations?

76 We agree that there is some seasonality in the AOD data, although not very strong. We also agree that it is

77 more appropriate to remove the seasonal cycle before computing correlations. As suggested, we have

**78** recalculated the correlation coefficients after removing the seasonal cycles with monthly means. The new

correlations are slightly smaller as we would expect. The new values have been updated in the revisedmanuscript.

6) Section 3.2. Diurnal cycle of winds should be better subordinated also with info on wind directions(given the paper focus).

- 83 Thank you for the comment. We believe that we have discussed the wind directions in detail in section
- 84 3.4 (old version). Figures 13-15 (old version) and relevant discussion describe the prevailing wind
- 85 direction in different seasons and at different altitudes. We have now revised this section with some
- additional information on breeze formation. We have also revised Figure 9 (old version) description to
- 87 clarify the relevance of wind directions in dust transport to our site further.
- (7) lines 495-497 analysis of day/night profiles. This para needs edits, as it stands it is very difficult to
   handle
- 90 Thank you for pointing this out. We agree that this paragraph was a bit unclear. As suggested, we have
- now substantially revised this paragraph to make our points on the cause of elevated dust loading at 1 2km in daytime and 5-7km in nighttime clearer.
- 93 (8) Fig 11 see comment (6) on wind directions
- 95 We believe that we have already addressed this point earlier. See the response to comment 6.
- 96

94

(9) lines 694+ the interaction of sea breezes with the Harmattan winds is explained in a very wordy andcontradictory manner, the para in a whole needs edits

- 99 Thanks for raising an interesting point. We agree it is important to discuss the interaction of harmattan100 winds with land/sea breezes, along with their effect on dust. Considering your suggestion, we have added
- following description in the end of section 3.2.3.:
- 102 "When the dust-laden harmattan winds arrive at the Red Sea coast, they encounter the land or sea breezes

depending upon the time of arrival, as discussed further in section 4.3. When they meet with the opposite

- sea breeze flow, the air mass rises up, bringing the dust to the upper levels. Such higher intrusion of dust is evident in the KAUST-MPL data (Figure 9, left) in the afternoon, during which the sea breezes are
- most active. The suspended dust is still visible in the upper levels (~2-3 km) in the night of August 10,
- 107 because the dust particles have not been deposited yet."
- (10) Fig 15 change please the arrow scale to see the differences in magnitude in the panels. Also you
   might wish to use a fine resolution for plotting wind arrows.
- 110 We understand your suggestion to change the scale of arrows in each figure panel for clarity. However,
- 111 we would like to use a consistent arrow scale in all the panels to show the difference in winds at different 112 altitudes and seasons. Therefore, we decided to keep the wind scale as it is.
- (11) Fig 17 and text. What is plotted is MSLP I guess, not surface pressure. Also consider using contoursfor SLP, as the color is not effective for identifying circulation patterns.
- We considered using contours but contours would reduce the cleanliness of the figure because of theborderlines and the contour numbers. We changed the naming from 'surface pressure' to mean sea level
- 117 pressure (MSLP) as suggested.
- (12) conclusive bullets should be grouped according to the paper flow. Otherwise, they are notconvincing, see also general comment.
- Thank you for this helpful suggestion. We have now grouped the conclusion in four headings accordingto paper flow, as suggested, and separated the discussion.

**123 **Response to reviewer #2**

- 124 This paper presents an approach coupling WRF-Chem, vertical profiles of a MPL lidar and photometric
- 125 measurements (AERONET) to study aerosols on the western coast of the Arabian Peninsula during 2015.
- 126 The authors also use MODIS, SEVIRI and CALIOP spaceborne observations to help them in their
- 127 interpretations. The authors aim to better understand the role of coastal breezes on the vertical distribution
- 128 of dust aerosols and assess the accuracy of the modelling compared to the observations.
- 129 This work is of scientific interest in the sense that the role of breezes and their interaction with the general
- 130 circulation of the atmosphere is not necessarily well evaluated at key locations on the planet, like in the
- 131 case of the region considered in this article. Dust aerosols are now recognized as having a significant role
- 132 on the radiative balance of some regions of the globe, but also on economic life (IPCC). This article is 133 therefore interesting, and the results of this research deserve to be published after major revision.
- increase increase in the results of this research deserve to be published after major revision.

**134 We are grateful to the reviewer for providing encouraging feedback on our manuscript.**

- 135 This article should be seriously revised and better organized before publication. There is a lot of
- repetitions throughout the text, which makes reading the article considerably more cumbersome and
- detracts from highlighting the main ideas. There is a need to group together the elements of discussion
- spread throughout the various sections. It is also necessary to be clearer about the objectives as this article
- 139 can be seen as a publication on the validation of WRF-Chem on the one hand and on the other hand it 140 claims an annual study on aerosols above the experimental site. The part on the cross-comparison
- between instruments and model should be well separated from the scientific interpretations. A "Model
- validation" section should be done more directly. This study is not conducted over a sufficiently long
- 143 period of time to be able to speak about climatology. It should therefore be repositioned in a more global
- 144 context to better highlight its scope. A major event has been observed and is the subject of a "case study",
- but is this event common in other years? Are the observed dust aerosol contents and their vertical
- distribution throughout 2015 reportable for other years? The discussion section is confusing and needs to
- 147 be better organized by a new structure of the article. It would be preferable to separate it from the
- 148 conclusion, which will then more clearly highlight the major findings of this study.

149 We agree that our manuscript needs to be better organized. We have extensively revised the paper in

- 150 different sections to reflect the reviewer's comments. To clarify the presentation, we added a separate
- section 'Model Validation' by combining section 3.1 (comparison of AOD and aerosol volume
- concentrations), parts of section 3.3 (diurnal cycle comparison of model winds with observation) and
   section 3.5 (case study). We have also added a new section for discussion with subheadings according to
- the flow of the paper.
- Regarding the "climatology", we agree with the reviewer's comment. In the revised manuscript, we have avoided using this term. The dust case study, we consider in the paper, is a typical recurring summer time dust event; so it is common in other years. Regarding the possible interannual variability, following the reviewer's suggestion, we have looked at the aerosol profiles for 2015 and 2016 separately. There is some interannual variability, as expected but it is relatively weak. We have now added the following text in the manuscript to clarify this issue:
- 161 "We observed some interannual variability while comparing the vertical profiles for 2015 and 2016, but was not too significant (Fig. S9). Therefore, the observed vertical distribution of dust aerosols can be
- 163 considered 'typical' for our region and possibly for other land-ocean boundaries (e.g., Rasch et al., 2001).

This is understandable because the synoptic winds causing large-scale dust events, and the diurnal-scale breezes that affect the dust distribution, both have strong seasonality over the study region (Kalenderski and Stenchikov, 2016; Parajuli et al., 2019). However, as demonstrated by our results, vertical profiles of aerosols can be affected by regional processes such as breezes, which indicate that the profiles can differ across different regions. Therefore, it is vital to examine the aerosol vertical profiles of a region to understand the regional climate."

109 understand the regional enmate.

170 Figure S9 is presented below.

L63. The vertical distribution of aerosols has been studied for decades using lidar measurements from the
ground-based, aircrafts, and satellites (LITE, CALIOP, GLAS) platforms. It is indeed an important

175 parameter for the assessment of the climatic impact of aerosols. Numerous publications exist. For

deontological reasons, I prefer to let the authors make their complementary bibliography, without

influencing them. They can research what has been done during INDOEX, ACE-2 or AMMA at theinternational level and elsewhere.

179 We are aware that our study is far not the first employing lidar observations for dust profile analysis.

However, there are no other such studies in the Red Sea coast region. Our research is essential becausethe presence of breezes over the Red Sea coast affects aerosols' vertical distribution.

We thank the reviewer for providing us examples of previous studies on vertical aerosol profiles using
 LIDAR data. We have compiled an extensive review of studies using LIDAR from satellites, field
 experiments, and networks, as suggested. This review is included in the revised manuscript, in the

185 introduction section:

186 "The vertical distribution of aerosols in the atmosphere has been studied for decades using LIDAR

measurements from several ground-based sites, aircraft, and satellite platforms, covering different regionsacross the globe. Several satellites are equipped with LIDAR to measure the vertical distribution of

189 aerosols. Lidar In-space Technology Experiment (LITE) was the first space lidar launched by NASA in 1994 onboard the Space Shuttle, providing a quick snapshot of aerosols and clouds in the atmosphere on a 190 global scale (Winker et al., 1996). LITE was followed by the Geoscience Laser Altimeter System (GLAS) 191 192 containing a 532-nm LIDAR, as part of the Ice, Cloud and Land Elevation Satellite (ICESat) mission, 193 which covered the polar regions (Abshire et al., 2005). Cloud-Aerosol Lidar with Orthogonal Polarization 194 (CALIOP) onboard CALIPSO (Cloud-Aerosol Lidar and Infrared Pathfinder Satellite Observations) 195 currently observes aerosol and clouds globally during both the day and night portion of the orbit with a 16-day repeat cycle since 2006 (Winker et al., 2013). Apart from satellites, several field experiments have 196 197 also been conducted using LIDAR to measure the vertical distribution of aerosols. The Indian Ocean 198 Experiment (INDOEX) field campaign (Collins et al., 2001; Rasch et al., 2001; Welton et al., 2002b) took 199 place in 1999 over the Indian Ocean, Arabian Sea, and the Bay of Bengal, in which an MPL system together with other instruments measured aerosol distribution in the troposphere. Similarly, an MPL 200 system was employed in the Second Aerosol Characterization Experiment (ACE-2) in 1997 over 201 202 Tenerife, Canary Islands, to understand the vertical distribution of dust/aerosols transported from North Africa and Europe to the Atlantic Ocean (Welton et al., 2000; Ansmann et al., 2002). African Monsoon 203 204 Multidisciplinary Analysis (AMMA), one of the largest international projects ever carried out in Africa, also measured aerosol vertical distribution using multiple LIDAR systems for a short period in 2006 205 (Heese and Wiegner, 2008; Lebel et al., 2010). Currently, several other coordinated LIDAR networks are 206 207 operating in different regions. They include the European Aerosol Research Lidar Network EARLINET (Pappalardo et al., 2014), German Aerosol Lidar Network (Boesenberg et al., 2001), the Latin American 208 Lidar Network LALINET (Guerrero-Rascado et al., 2016), the Asian dust and aerosol lidar observation 209 210 network AD-Net (Shimizu et al., 2016), and the Commonwealth of Independent States Lidar Network 211 CIS-LiNet (Chaikovsky et al., 2006)."

- 212
- L92. Clouds necessarily influence the lidar inversion which usually requires a reference, usuallymolecular in the upper troposphere. Can you clarify your statement?
- 215
- **216** We agree with the comment. We have clarified the text to highlight the benefit of LIDARs data as

compared to the passive satellite sensors, which are generally based on visible bands. We have also added
 appropriate references in the revised text:

- 219 "AERONET stations and passive satellite sensors are further limited because they cannot retrieve aerosol
- 220 properties during the night. LIDARs help to overcome these limitations because they provide high-

221 frequency measurements, even at night. Furthermore, LIDAR signals can penetrate thin and multilayer

clouds, which are usually overlooked by passive satellite sensors (Winker et al., 1996; Winker et al.,

2009), thus improving the detection of aerosol layers at different altitudes. Therefore, LIDAR data are
 essential for understanding the diurnal variability of aerosols and their climatic effect."

- Sub-section 2.1. The scheme on the breeze would be better placed in the revised discussion.
- 226 Agreed. We have now moved the figure to the revised discussion section.
- 227 L157. Use "annual study" rather than "climatology".

228 We agree that our study is not truly a climatological since we have only two-years of data. Therefore, we

have avoided using the term 'climatology' entirely in the revised manuscript. We now use throughout thetext the word 'seasonal average', with the averaging period given in bracket.

- 231 L163. The CALIOP instrument?
- 232 Corrected.
- 233 L173. Replace version by data?
- 234 Corrected.
- 235 L190. Define DOD.
- 236 Corrected as suggested.

L210 and following. How do you find the absorption coefficient with a MPL? More needs to be saidabout the implemental retrieval.

We agree that this statement was not clear. Considering your comment, we have revised the entire paragraph to clarify the implemented retrieval, which is reproduced below:

241 "The colocation of the KAUST-MPL and AERONET station provides an opportunity to get a more

comprehensive microphysical picture when the MPL data are combined with AERONET sun-photometer
 measurements. We employ GRASP (Generalized Retrieval of Aerosol and Surface Properties, Dubovik et

al., 2011, 2014), which is an open-source inversion code that combines different types of remote sensing
 measurements, such as radiometer and LIDAR observations, to generate fully consistent columnar and

vertical aerosol properties (Lopatin et al., 2013). We take aerosol characteristics from the AERONET

retrieval including size distribution, absorption, scattering optical depth, and refractive index. These

parameters serve as inputs to GRASP, together with MPL data, to generate height-resolved aerosol fields
 such as aerosol extinction, absorption, and mixing ratios."

L240 and following. Aren't there difficulties in parameterizing turbulence at such scales? Can you justifythe choice of the PBL scheme? This is an important element for this type of study.

After reading your comment, we realized that we had wrongly mentioned the PBL scheme that we had

used. We actually used YSU PBL scheme Yonsei University YSU (Hong, Noh, and Dudhia, 2006) not

254 MYJ (Janjic, 1994) as mentioned. We have updated this information in the revised manuscript. As

suggested, we have added the following justification for the use of PBL scheme in the revised manuscript:

256 "Several studies compare the performance of PBL schemes in WRF, showing mixed results under

different model settings (e.g., Saide et al., 2011; Fountoukis et al., 2018; Fekih and Mohamed, 2019).

258 However, these studies have not directly compared the aerosol vertical profiles. Preliminary results

showed that the choice of the PBL parameterization did not have a significant impact on the vertical
distribution of aerosols in our case. In our simulations, we use the YSU PBL scheme, which is one of the

most commonly used schemes, as suggested in the literature (e.g., Fountoukis et al., 2018; Fekih and

262 Mohamed 2019)."

Sub-grid turbulence is not resolved in our simulations, but is parameterized. We agree that reproducingthe turbulence effects is challenging and can be improved in WRF-Chem. However, such exercise is out

- 265 of scope of this study.
- 266 L245. Define MENA.
- 267 Done.
- L248. Remind the definitions of u and v.

- 269 Definitions added.
- 270 L300-306. This approach assumes that there is no internal mixing.
- 271 Yes, we agree.
- 272 L326. Climatology?
- 273 We have avoided using this term throughout the manuscript.
- L343. Give the equation.
- Correlation coefficient and mean bias error are fairly used terms so we chose not to provide the equationfor brevity of the manuscript.
- L350. The "robust" term is somewhat strong with correlations between 0.6 and 0.7.
- 278 We agree. We replaced the term 'robust' with 'reasonable'.
- 279 L393-398. Example of duplication.
- 280 This is an important suggestion. We have revised and moved these lines to the section 2.1, Study site.
- 281 Figure 6a. Define WS. Climatology?
- 282 Corrected as suggested.
- 283 L403-406. Already mentioned.
- 284 We have moved this to section 2.1 and edited to avoid repetition.
- 285 L407-417. Combine with what was already mentioned on the sea breeze.
- 286 The section has been revised as suggested.
- L432-433. No, CALIOP inversions use a lookup table with backscatter, color ratio and depolarization asinputs.
- 289 We agree that this statement was not clear. Here, we do not want to provide too much details of CALIOP
- algorithm; we only want to differentiate CALIOP algorithm in terms of lidar ratio as compared to MPL.We have revised the sentence and the whole paragraph for clarity, as presented below:
- 292 "The difference in vertical profiles retrieved from KAUST-MPL and CALIOP data could be related to the
- differences in the algorithm and resolution between the two datasets. Firstly, while retrieving aerosol
   extinction profiles, the CALIOP algorithm uses different prescribed extinction-to-backscatter (lidar ratio)
- for a set of aerosol types from a lookup table (Omar et al., 2009; Winker et al., 2009; Kim et al., 2018). In
- addition, the CALIOP algorithm has difficulty in identifying the base of aerosol layers accurately. In
- 297 particular, the level-3 algorithm ignores the 'clear air' between the surface and the lowest aerosol layer
- when averaging to avoid underestimation of extinction in the lower part of the aerosol profile (Winker et
- al, 2013). In contrast, the MPL algorithm assumes an averaged lidar ratio for the whole column based onthe aerosol PSD, refractive index, and sphericity, in such a way that it satisfies both AERONET and MPL
- 301 co-incident data. Because of the assumption of a constant lidar ratio, MPL retrievals near the surface
- 302 could be erroneous, especially when multiple aerosol layers are present (Welton et al., 2002a). Secondly,
- 303 KAUST-MPL is a point measurement that captures the temporal evolution of the dust storms better than
- 304 CALIOP because it has a higher temporal resolution. For instance, CALIOP can undersample or overlook

- some dust events that last only for a few hours. On the other hand, CALIOP could sample more spatial 305
- 306 details of a dust storm because of its extended coverage along its track compared to KAUST-MPL data.
- Nonetheless, these two datasets complement one another, and their combined use can be beneficial in 307
- understanding the large-scale dust storms." 308
- 309 L440-441. Be careful because the distance between two ground tracks is large.
- 310 We understand your point. We only say this in comparison to point measurements such as LIDAR. We 311 have revised this statement (see the response to the previous comment).
- 312 Fig. 7. Height is the altitude a.g.l.?
- 313 It is above sea level, we added this information in the caption.

314 Fig. 7c (MPL during nighttime) and related discussion. What we see above 5 km looks like contamination

- by semi-transparent clouds (or an average with cloudy profiles). This may also be why there is such a 315 316 large discrepancy with the model.

317 We have taken your comment on possible cloud contamination in our retrievals seriously. We have added

- 318 discussion in several places to make the readers aware of this issue. Further, we have done some
- 319 calculations using depolarization to check such a possibility. We have added relevant discussion mainly in two places: 'data and methods' and another in 'discussion.' The following text is added in the data and 320
- 321 methods section to clarify this issue:

322 "The GRASP algorithm relies on an external cloud masking. Overnight lidar retrievals are performed 323 only when cloud-free AERONET sun-photometric observations are available in the preceding evening

- and following morning. The AERONET cloud-masking algorithm is considered the golden standard, 324
- 325 providing very reliable filtering of thick and broken clouds (Holben et al., 1998). In this regard, only
- clouds that form specifically at night and are undetectable by sun-photometric observations in the evening 326
- 327 and morning could influence our retrieved extinction profiles. At the same time, retrieval of these profiles,
- 328 to a large extent, relies on detailed columnar aerosol properties retrieved before and after nighttime
- 329 observations. An attempt to retrieve cloudy profiles under the assumption of cloud-free aerosol columnar
- properties should result in higher fitting errors, and therefore should be easily detectable." 330
- 331 The following text is added in the discussion:

"To better understand the origin of two elevated dust layers observed (~1-2 and 6-7 km) and investigate 332 333 the possibility of thin-cloud contamination in our MPL retrievals, we analyzed the volume-depolarization 334 profiles provided by the KAUST-MPL, synchronous to the attenuated backscatter profiles used in the 335 retrievals. The average volume depolarization value in the lower atmosphere (1-2 km) was estimated to be 336 13-14% on average and 7-8% for the upper part (6-7 km) for the selected period. Such values indicate that 337 high extinction values in this altitude range cannot come exclusively from clouds because pure water clouds generally yield a 1-2% depolarization value and ~30% or even higher in the case of cirrus clouds 338 339 (e.g., Del Guasta and Valar, 2003). The lower depolarization value in the upper part could be explained by the fact that the aerosol particle sizes are much finer than those in the lower part. At the same time, a 340 341 lower depolarization value also suggests the possibility of partial influence by thin clouds. The presence 342 of thin clouds can probably cause some overestimation of aerosol concentrations and extinction at these 343 altitudes. However, such an overestimation is expected to increase the fitting errors, which are easily 344 detectable, as mentioned earlier. To ascertain this with full confidence, we plan a further analysis utilizing simultaneous retrieval of sun-photometric observations together with backscatter and volume-345

depolarization profiles provided by KAUST-MPL in the future." 346

- 347 L458-460. The difficulty in retrieve aerosols close to the surface is not the same for CALIOP and the 348 MPL.
- Agreed. We removed the statement here and clarified this in the discussion section added; please refer to 349 our response in the earlier comment. 350
- 351 L468. I do not think it is very good in the spring when the model gives much higher values.
- 352 Agreed. We meant to say both summer and spring. We revised the text accordingly.
- 353 Sub-section L461. The model does not mark the PBL top well and it gives much higher aerosol extinction
- 354 coefficients. It would be interesting to see the temporal evolutions of the PBL height deduced from the 355 MPL and the numerical scheme chosen for WRF. A good representation of the PBL is fundamental to
- take into account the PBL/free troposphere exchanges. Moreover, to compare WRF and the MPL, it 356
- would be more interesting to have an OSSE (observing system simulation experiment) as for example in 357 Wang et al. (ACP, 2014). 358
- 359 We have presented the model PBLH in the supplementary information (Figure S5 in the revised
- supplement), which varies greatly in different seasons. In the MPL profiles, we can assume that the PBL 360
- height coincides with the top of an aerosol layers. This height is consistent in all datasets (~7km) with 361
- 362 minor discrepancies (Figure 7/8 in the old version). Therefore, we believe that the model calculates the
- 363 PBL height quite reasonably. Thank you for the suggestion regarding OSSE. We agree that such analysis 364
- are meaningful, however, such a detailed analysis is out of scope of this study.
- 365 L491-497. I do not understand what is being demonstrated here. Dust aerosol layers are often above the PBL and in coastal areas the PBL is lower. 366
- 367 We agree that this section is confusing as also pointed by other reviewers. We have now revised this section to make our point clearer on the origin of elevated dust loading observed at two heights, one at 1-2 368 km and another at 6-7 km. We have also modified the text in several other places to connect this concept 369 370 with the examples. We meant to say that the particular shape of aerosol profile that we observed is similar 371 to the profiles observed in some other studies during 'dust storms'. We hope this is clear now in the revised text. Please read the last paragraph in the revised text presented in response to the comment on 372 373 L519 below.
- 374 L504-505. Beware of cloud signatures on lidar profiles.
- 375
- 376 With additional information on possible cloud contamination provided earlier, we believe this is clear 377 now
- 378 L502-503. Aerosols emitted non-locally are most often transported at higher altitudes, above the PBL. 379 This is therefore not an exceptional case.
- 380 Agreed. The text is adjusted accordingly.
- Sub-section L519. I do not understand what the "clear day"/"dusty day" comparison brings to the 381 understanding of the differences between MPL and model. When there is no dust, it is normal that we do 382
- not see anything, it doesn't prove anything. 383
- We agree that these paragraphs were not clear as also pointed by reviewer #1. We have revised it to 384 convey our intended message, as we are not talking about the difference between MPL and model. Our 385

analysis is to uncover what would be the origin of the elevated dust layer observed in MPL. The revisedparagraphs are given below:

"To understand the causes of the elevated dust maxima in the KAUST-MPL profiles at ~1-2 km altitude
in the daytime and 5.5-7 km in the nighttime, we separately analyzed the profiles under a clear sky and
dusty conditions. We define 'clear days' as the days with a daily mean of AOD at KAUST less than 0.25
and 'dusty days' as the days having daily-mean AOD greater than 0.75, using either MODIS AOD or

**392** AERONET AOD to maximize data availability during large-scale dust events.

Figure 12 shows the average extinction profiles for clear and dusty conditions from KAUST–MPL datafor 2015/16 obtained using the above criteria. The daytime profile (Fig. 12, left) shows a similarly

elevated dust loading at 1-2 km height, as noted earlier in Figures 10/11, but is much more prominent.

396 Since 'dusty days' correspond to very high AOD conditions (AOD>0.75) expected during dust storms, we

397 can infer that the observed elevated dust loading at 1-2 km corresponds to large-scale dust storms. Studies

398 have shown that this shape is characteristic of dust profiles observed during large-scale dust events near

and-ocean boundaries (Khan et al., 2015; Senghor et al., 2017). Marenco et al. (2018) also observed a

400 similarly elevated dust loading over the eastern Atlantic at a comparable height in their airplane

401 observations during the 'heavy dust' period.